# A diversity of novel type-2 innate lymphoid cell subpopulations revealed during tumour expansion

Clara Wenjing Xia[1,2,3,4,5,9], Iryna Saranchova[1,2,3,4,5,6,7,8,9], Pablo L. Finkel[1,2,3,4,5], Stephanie Besoiu[1,2,3,4,5], Lonna Munro[1,2,3,4,5,6,7,8], Cheryl G. Pfeifer[1,2,3,4,5,6,7,8], Anne Haegert[2,8], Yen-Yi Lin[2,8], Stéphane Le Bihan[2,8], Colin Collins[2,8] & Wilfred A. Jefferies [1,2,3,4,5,6,7,8✉]

Type 2 innate lymphoid cells (ILC2s) perform vital functions in orchestrating humoral immune responses, facilitating tissue remodelling, and ensuring tissue homeostasis. Additionally, in a role that has garnered considerably less attention, ILC2s can also enhance Th1-related cytolytic T lymphocyte immune responses against tumours. Studies have thus far generally failed to address the mystery of how one ILC2 cell-type can participate in a multiplicity of functions. Here we utilized single cell RNA sequencing analysis to create the first comprehensive atlas of naïve and tumour-associated lung ILC2s and discover multiple unique subtypes of ILC2s equipped with developmental gene programs that become skewed during tumour expansion favouring inflammation, antigen processing, immunological memory and Th1-related anti-tumour CTL responses. The discovery of these new subtypes of ILC2s challenges current paradigms of ILC2 biology and provides an explanation for their diversity of function.

[1] Michael Smith Laboratories, University of British Columbia, 2185 East Mall, Vancouver, BC V6T 1Z4, Canada. [2] The Laboratory for Advanced Genome Analysis (LAGA), The Vancouver Prostate Centre, Vancouver General Hospital, 2660 Oak Street, Vancouver, BC V6H 3Z6, Canada. [3] Department of Microbiology and Immunology, University of British Columbia, 2350 Health Sciences Mall, Vancouver, BC V6T 1Z4, Canada. [4] Centre for Blood Research, University of British Columbia, 2350 Health Sciences Mall, Vancouver, BC V6T 1Z4, Canada. [5] Department of Zoology, University of British Columbia, 6270 University Blvd., Vancouver, BC V6T 1Z4, Canada. [6] The Djavad Mowafaghian Centre for Brain Health, University of British Columbia, 2215 Wesbrook Mall, Vancouver, BC V6T 1Z4, Canada. [7] Department of Medical Genetics, University of British Columbia, 2350 Health Sciences Mall, Vancouver, BC V6T 1Z4, Canada. [8] Department of Urologic Sciences, University of British Columbia, Vancouver, BC V5Z 1M9, Canada. [9] These authors contributed equally: Clara Wenjing Xia, Iryna Saranchova. ✉email: wilf@msl.ubc.ca

Physiologically, tumour-infiltrating leukocytes (TILs) can respond to transformed cells at an early stage of cancer development[1]. Tumour cells can be recognized by TILs, such as natural killer (NK), natural killer T (NKT), and γδ T cells, which produce large amounts of IFN-γ that drive cellular immunity and subsequently, stimulation of anti-tumour immune-responses. As a consequence, NK cells and cytotoxic T lymphocytes (CTLs) can directly kill and specifically eliminate tumour cells[1–4]. Countering the positive aspects of immunity to tumours, the selective pressure of immune surveillance by CTLs on genetically unstable tumour populations may yield tumours that have lost expression of antigen processing machinery (APM) components, often resulting in reduced assembly of functional major histocompatibility complex (MHC) molecules that would normally bind peptide fragments derived from pathogens and tumours and display them on the cell surface for recognition by the appropriate CTL[1,5–8]. Several types of cancer, including breast cancer[9,10], renal carcinoma[11], melanoma[12–14], colorectal carcinoma[15], head and neck squamous cell cancer[16], cervical cancer[17], lung carcinoma[18,19] and prostate carcinoma[20–22], exhibit APM deficits and show a clear correlation between human leukocyte antigen (HLA) downregulation and poor prognosis[20–25]. Depending on the tumour type, APM component loss and MHC-I (HLA-I) molecules with an immune escape phenotype may be observed in up to 90% of patients, which can then lead to tumour invasiveness and increased metastatic potential[18,20–25]. Under these circumstances, tumours become 'invisible' or unrecognizable by CTLs and may also become refractory to emerging immunotherapeutics[20–22]. However, this can be overcome by complementation with APM genes or by small molecule inducers of APM genes[1,6–8,14,18,19,26,27].

Overcoming immune evasion mechanisms is crucial to address tumour resistance and unlock the full potential of immunotherapy. Furthermore, the limited success of chimeric antigen receptor (CAR) T cell therapies for solid tumours has spurred an exploration of alternative immune cells in cell-based immunotherapies[28,29]. Simultaneously, there is an urgent demand for advanced methods to identify responsive patients[30]. In addition, immune dysregulation caused by chronic inflammation can also lead to modified expression of immune checkpoint inhibitors and alter the transiting and effector functions of TILs, which may instead promote tumour development. Examples of immune interference that lead to tumour development and progression are the secretion of IL-10 and the expression of PD-L1 and CTLA-4 by regulatory T cells (Tregs) to suppress CTL functions[31] and the expression of inhibitory receptors on T cells, such as PD-1 and CTLA-4, to inactivate effector T cells[32]. Compounding this, a new regulatory subpopulation of innate lymphoid cells (ILCs) named ILCregs, which expresses *Id3* and produces IL-10, has recently been shown to modulate the inflammatory response[33–35] and are thought to contribute to cancer progression and therefore, must also be overcome to empower cancer immunotherapies[36].

Our understanding of the role of ILCs in cancer immunity is still limited. ILCs integrate innate and adaptive immune responses and control numerous physiological processes[37] by producing cytokines in response to danger signals. It has been generally accepted that ILCs do not express a functional B or T cell receptor and, therefore, do not mediate an antigen-specific response[38]. Instead, the functional specificity and classification of ILCs are based on their transcription factor and cytokine expression patterns. Currently, three defined subsets of ILCs exist, namely types 1, 2, and 3[39–42]. ILC1s are T-bet+ pro-inflammatory cells that produce large amounts of TNF-α and IFN-γ. ILC2s are GATA-3+ cells, which are developmentally and functionally dependent on the presence of the interleukin-33 (IL-33) protein in order to secrete $T_H2$-type cytokines, such as IL-4, IL-5, IL-9, IL-13, and signaling for Th2 adaptive

responses[37]. ILC3s have been associated with the expression of transcription factor RORγt and are involved in lymphoid tissue release of IL-17, IL-22) and IL-23[43].

Until recently, ILC2s[44,45] were solely considered to contribute Th-2 cytokines, enhancing Th-2 responses in conditions such as asthma[37]. In this context, ILC2s were traditionally viewed as the most homogenous and stable subgroup within the ILCs. However, recent studies are revealing the existence of novel ILC2 subpopulations[46–48]. The flagship study on ILC2 heterogeneity by Huang et al. identified a second subset of ILC2s that responds primarily to IL-25 instead of IL-33[49]. These inflammatory ILC2s appear undetectable in steady state but can be elicited upon alarmin release and can differentiate into IL-33-responding ILC2s, or can upregulate RORγt and acquire ILC3 functions. This is accompanied by an increase in IL-17 secretion and what appears to be central in ILC2 plasticity: a halt in the production of IL-5 and IL-13[47]. IL-1β is a potent ILC2 activator that upregulates IL-5, IL-13, IL-33R (ST2) and IL-25R[50,51], and interestingly, ILC2-to-ILC1 plasticity was later reported[52–54], where IL-1β was found to promote T-bet and IL-12Rβ2 expression which, in response to IL-12 and an accompanying inflammatory environment, can convert ILC2s into IFN-γ-producing ILC1s. In addition, ILC2 MHC-II and CD80 molecules are increased in response to IL-1β, suggesting elevated ILC2 and T cell interactions. Other ILC2-subpopulations have been discovered in non-cancer settings[22,43,55–61].

ILC2s are thought to perform a paradoxical role in cancer. On one hand, there is now considerable evidence implicating ILC2s in the tolerogenic processes related to tumour immune evasion, and produce large amounts of IL-13, IL-5, and IL-4. Furthermore, some studies relate ILC2s to poor cancer prognosis[62,63]. IL-13 production has been recently linked to the activation of myeloid-derived suppressor cells (MDSCs) and generates an ILC2-MDSC immunosuppressive axis that may facilitate tumour immune escape[64]. This skewing toward type 2 immunity and active MDSC recruitment has led to recurrence of bladder cancer in humans[65]. In contrast, challenging this paradigm, an exploration into how cancer cells evade the host immune response and become metastatic has led to the discovery of a novel function for ILC2s in promoting Th1 responses, and thereby reducing tumour growth and metastasis[20–22]. We found that the presence of IL-33 in the microenvironment simultaneously stimulates both the MHC-I expression and antigen processing in epithelial cells, as well as the number and functional activities of ILC2s, suggesting the co-regulation of these processes[20–22]. Therefore, reduced IL-33 expression in metastatic mouse carcinomas simultaneously decreases both the MHC-I expression and the functional activities of ILC2s and provides a previously undescribed mechanism of immune escape for tumour cells. Moreover, we demonstrated a similar association between IL-33 and HLA expression in clinical specimens at different stages of human prostate tumour development[20–22]. Our work also documented that the presence of ILC2s significantly inhibits tumour formation in wild-type chimeric mice bearing tumours expressing IL-33 when compared to RORα-/- chimeras, which lack ILC2s. In comparing RORα-/- to wild-type controls, we have observed increased spread of circulating tumour cells and formation of metastases in distal organs[20–22]. We have also found that co-culturing ILC2s and CD8+ T cells with metastatic tumour cells enables metastatic cells to overcome the antigen presentation deficiencies and heighten the CTL effector mechanisms[20]. ILC2 identity in the context of cancer remains largely undescribed beyond our initial discovery[20], however, recently, two reports have emerged implicating ILC2s in tissue-specific cancer immunity. Following on from our studies, Moral et al. report that ILC2s

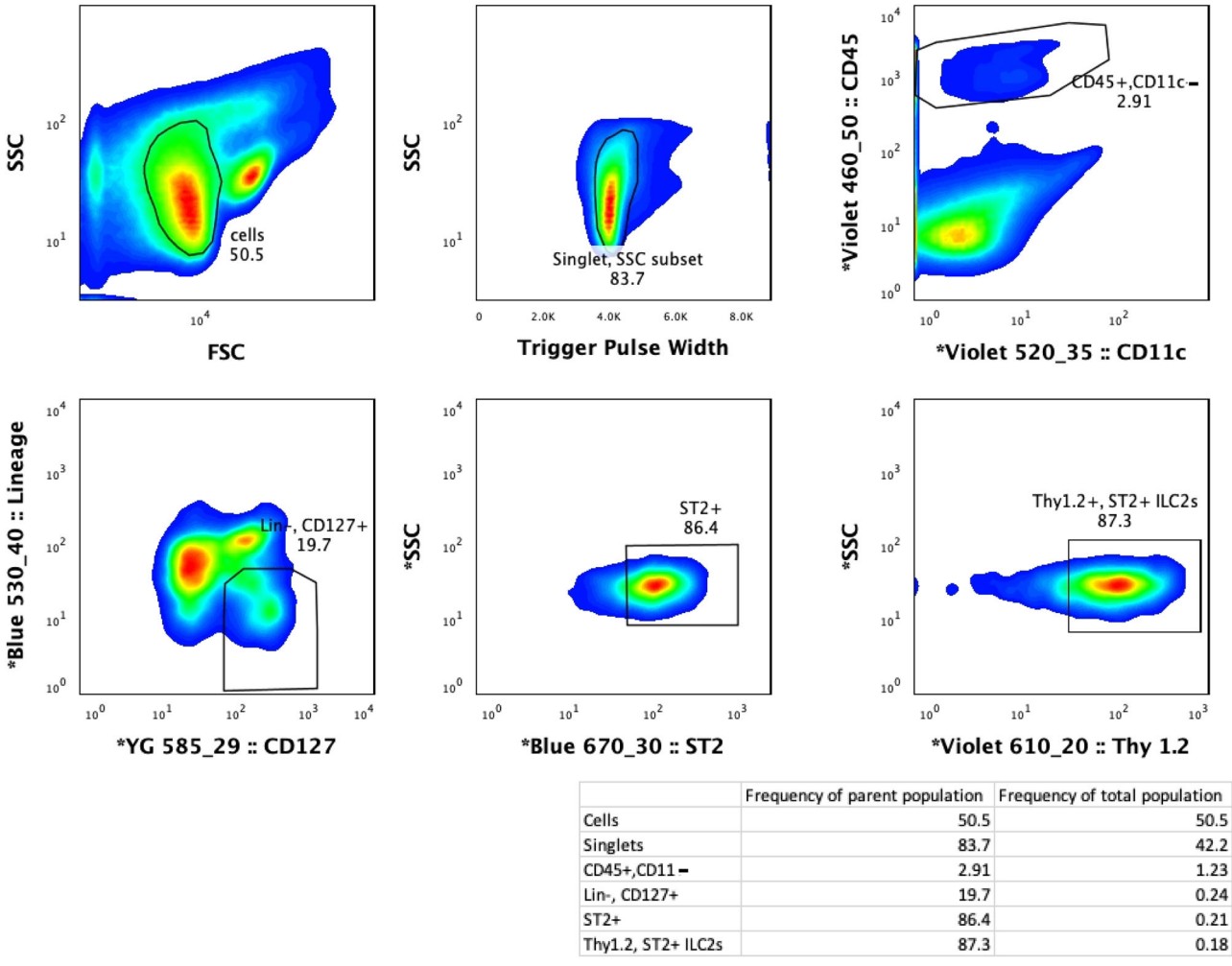

| | Frequency of parent population | Frequency of total population |
|---|---|---|
| Cells | 50.5 | 50.5 |
| Singlets | 83.7 | 42.2 |
| CD45+,CD11− | 2.91 | 1.23 |
| Lin-, CD127+ | 19.7 | 0.24 |
| ST2+ | 86.4 | 0.21 |
| Thy1.2, ST2+ ILC2s | 87.3 | 0.18 |

**Fig. 1 Gating strategy for isolation of ILC2s from lungs.** The gating of a standalone experiment to illustrate the gating strategy, which is the same for all ILC2 isolations in the study. ILC2 cells were enriched with EasySep Mouse ILC2 Enrichment Kit (STEMCELL Technologies) first then sorted from lungs of tumour bearing animals by FACS as Lin− ST2 + CD127 + Thy1.2+ cells. Sequential gating strategy is based on cell size [P1: small and non-granular cells, forward scatter (FSC) versus side scatter (SSC)], depletion of doublets (P2: SSC vs. Trigger Pulse Width), and selection of CD45 + cells. Cells with lineage-related markers were rigorously depleted during isolation process and the resulting cell numbers at each stage of purification is shown. The final gate (double positive for ST2 and Thy1.2) includes ILC2s isolated, enriched and sorted from lung tissue.

within pancreatic tumours express the checkpoint inhibitor PD-1 and that upon treatment with anti-PD-1 immunotherapy, these ILC2s expand, become activated, and augment anti-tumour immunity by priming CD8+ T cells[66]. Finally, Wang et al.[67] describe trans-differentiation of tumour-infiltrating ILCs (ILC1, 2, and 3) during progression of colorectal cancer that highlights the sensitivity, responsiveness and plasticity of ILCs during the growth of tumours[67].

Overall, the contradictory functional data on the role of ILC2s in cancer suggest the possibility that there may exist multiple, undescribed subpopulations of ILC2s. Here we seek to under-stand the underlying mechanisms to explain these apparently contradictory studies. Our aim was to investigate the hypothesis that ILC2s are highly adaptable and exist as various subsets in the context of tumour immune surveillance. Thus, we conducted a single-cell RNA sequencing (scRNA-seq) study to create the first atlas of ILC2 subpopulations elicited by tumours and found that proinflammatory Th1 ILC2s expand in response to the development and growth of tumours, thereby demonstrating a bidirectional interplay between ILC2 and the cancer microenvironment.

## Results

**ILC2s is a heterogeneous population with pro-inflammatory phenotypes and anti-tumoral potential.** To confirm that the presence of tumour in the organism induces the plasticity of ILC2 cells by changing their transcriptional profiles, we performed single cell RNA sequencing (scRNA-seq) analysis on ta-ILC2s isolated during tumour development and compared it to naïve ILC2s from mice without tumour nor IL33 activation treatment. All ILC2s are enriched from mouse lungs and labeled for fluorescence-activated cell sorting (FACS). The representative sorting strategy, used for all ILC2 isolations, is described in ref. 68 and illustrated in Fig. 1. In all cases, this purification method yielded ILC2s with a purity of greater than 99%[68]. To capture the complete heterogeneity regardless of time of sample collection and activation status, we first integrated all datasets including naïve, tumour-associated and activated (ta-activated), and ta-non-activated through "anchor" finding and integration. This aims to identify commonalities while enabling further compara-tive analysis. Eight clusters, namely C0-C7, were called based on single-cell transcriptome of the naïve and ta-ILC2s following the standard workflow of Seurat[69], signifying the heterogeneity in this

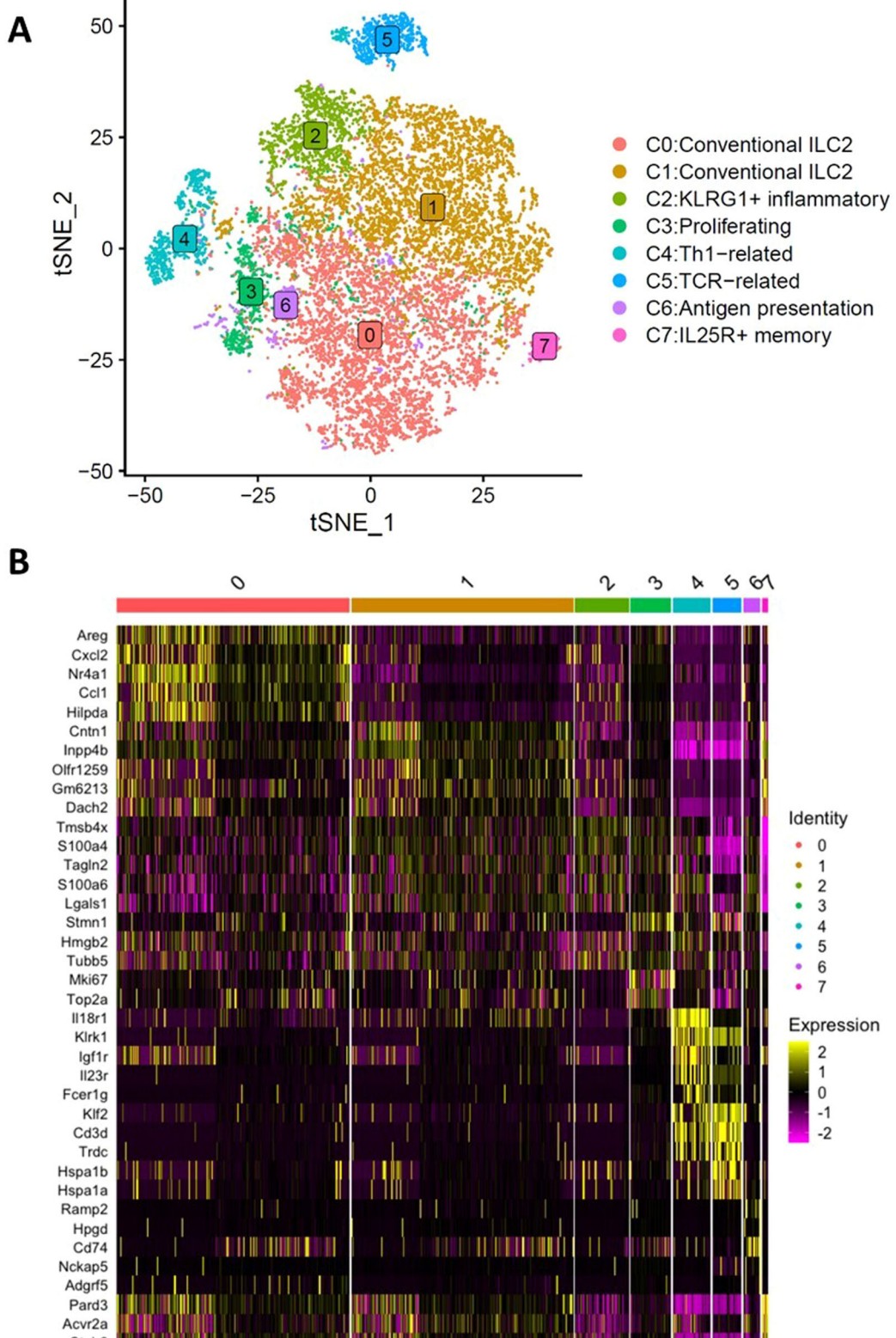

**Fig. 2 ta-ILC2s is a heterogeneous population with pro-inflammatory phenotypes, as a response to a developing neoplastic disease at week 2 and 3.**
**A** Graph-based clustering of ILC2s using Seurat R package. t-distributed stochastic neighbor embedding (tSNE) plots were generated to visualize graph-based clustering of cells. Both naïve and ta-ILC2s are merged in the analysis to reveal common subpopulations of ILC2s. **B** Heatmap with the top ten differentially expressed genes per cluster, ranked by log-transformed fold change in expression levels in each cluster compared with all other clusters. Heterogeneity is observed in the overall ILC2 population (ta-ILC2 and naive ILC2) with differentially expressed genes.

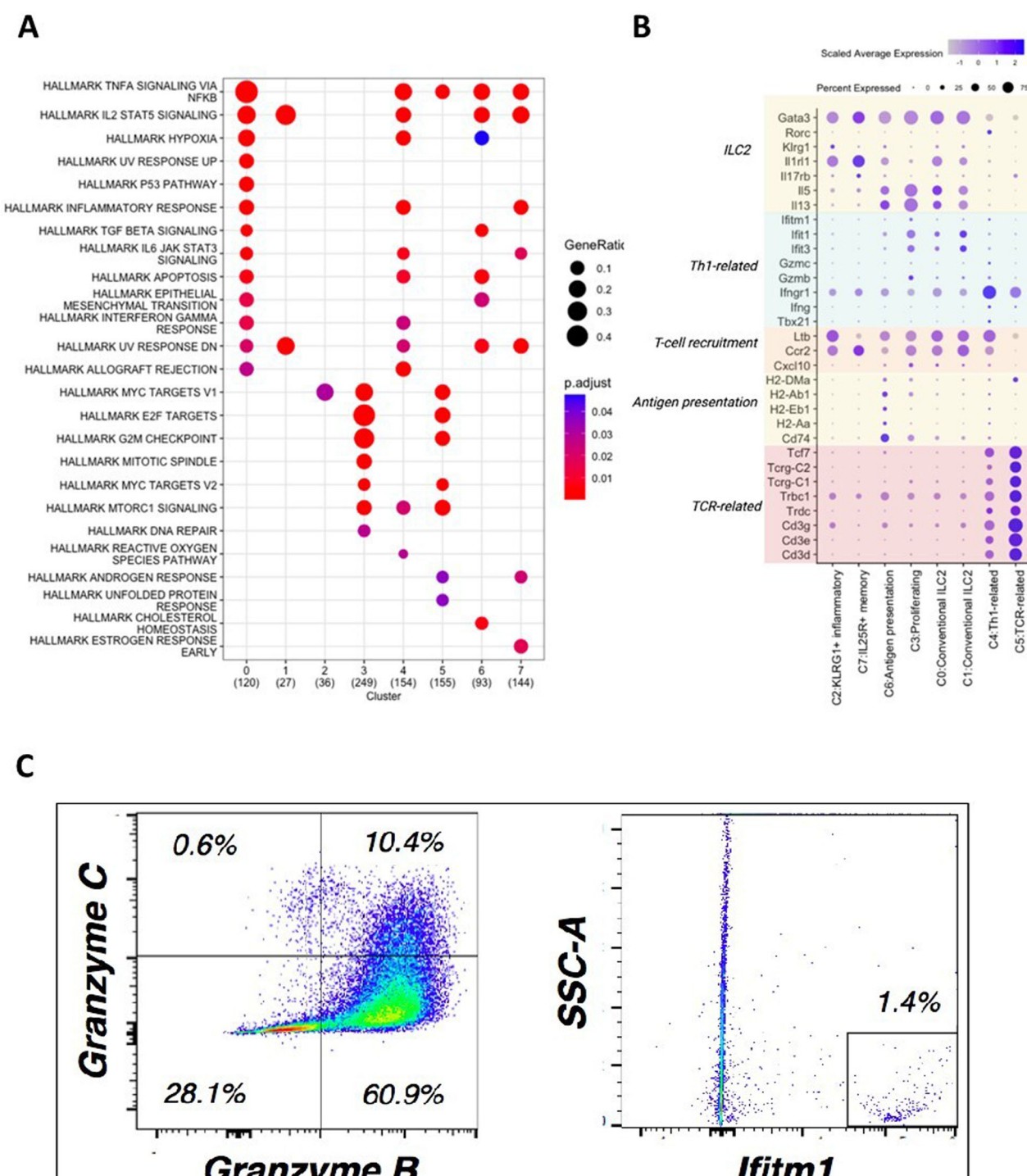

**Fig. 3 Subpopulations of ILC2s are enriched with different molecular signatures indicating diverse cellular functions. A** Dotplots indicate different gene sets being enriched in each subpopulation/cluster. Molecular Signature Database Hallmark gene sets are used as reference. Numbers in parenthesis indicate the number of overexpressed genes from each cluster that are overrepresented in the corresponding pathway. **B** Expression of selected ILC2 markers-of-interest (GOIs) are differentially expressed by different clusters. Dotplot shows marker-of-interest expression per cluster. Color intensity indicates log-scaled mean gene expression level. Dot size indicates the fraction of cells in the cluster for each gene. **C** Intracellular staining for granzymes B & C: a portion of ta-ILC2s are positive for both (10.4%), but mostly for granzyme B (60.9%). A fraction of ta-ILC2s is positive for intracellular IFITM1 (1.4%) confirmed by flow cytometry.

previously considered homogeneous type 2 effectors within the lung tissue (Fig. 2a).

Given the lack of canonical markers for subtypes of ILC2s in the existing literature, we aimed to elucidate the subtype identities of the clusters by identifying the marker gene sets that are differentially expressed within each of the clusters. We focused on the genes that are up-regulated in each of the clusters illustrated in the heatmap which shows the top 5 upregulated genes per cluster (Fig. 2b).

To examine the functionality of each cluster inferred by over-representation of their respective upregulated markers, we

**Table 1 Average expression of GOIs in naïve-, ta-non activated-, ta-activated ILC2 cells.**

| Pathway of Interest | GOI | Average Expression (total ILC2s) | | | Expression log2-FoldChange* (compared to all other samples) | | |
|---|---|---|---|---|---|---|---|
| | | naïve | ta-non-activated | ta-activated | naïve | ta-non-activated | ta-activated |
| ILC2 | *Gata3* | 7.625 | 3.457 | 8.766 | | | −0.057 |
| | *Rorc* | 0.093 | 0.050 | 0.021 | 0.049 | | |
| | *Klrg1* | 0.656 | 0.253 | 0.085 | 0.218 | −0.090 | −0.166 |
| | *Il1rl1* | 7.886 | 1.201 | 1.046 | 0.351 | −0.345 | −0.144 |
| | *Il17rb* | 0.523 | 0.205 | 0.005 | | | |
| | *Il5* | 1.589 | 7.035 | 8.635 | 0.366 | −0.192 | −0.258 |
| | *Il13* | 0.513 | 48.136 | 30.638 | −0.054 | −0.034 | 0.077 |
| Th1-related | *Ifitm1* | 0.033 | 0.040 | 2.024 | −0.643 | −0.483 | 0.883 |
| | *Ifit1* | 0.019 | 0.061 | 7.496 | −0.320 | −0.251 | 0.468 |
| | *Ifit3* | 0.004 | 0.012 | 4.410 | −0.253 | −0.199 | 0.374 |
| | *Gzmc* | 0.018 | 0.011 | 2.559 | | −0.058 | 0.098 |
| | *Gzmb* | 0.079 | 0.126 | 8.839 | −1.214 | −0.982 | 1.591 |
| | *Ifngr1* | 4.467 | 2.513 | 0.716 | | 0.194 | −0.474 |
| | *Ifng* | 0.087 | 0.145 | 0.024 | 0.030 | 0.030 | −0.051 |
| | *Tbx21* | 0.086 | 0.001 | 0.000 | | | |
| T cell recruitment | *Ltb* | 6.151 | 2.411 | 9.967 | 0.275 | −0.328 | |
| | *Ccr2* | 5.137 | 1.713 | 5.660 | | | |
| | *Cxcl10* | 0.211 | 0.470 | 3.815 | −0.749 | −0.506 | 0.983 |
| Antigen presentation | *H2-Dma* | 0.060 | 0.180 | 0.145 | −0.068 | 0.101 | |
| | *H2-Ab1* | 0.093 | 0.121 | 0.300 | | −0.028 | 0.052 |
| | *H2-Eb1* | 0.103 | 0.110 | 0.082 | 0.023 | | −0.024 |
| | *H2-Aa* | 0.113 | 0.109 | 0.085 | 0.038 | | −0.033 |
| | *Cd74* | 0.359 | 0.772 | 1.459 | −0.080 | −0.105 | 0.151 |
| TCR-related | *Tcf7* | 0.444 | 1.180 | 0.057 | −0.088 | 0.573 | −0.363 |
| | *Tcrg-C2* | 0.132 | 1.104 | 0.024 | 0.011 | 0.228 | −0.178 |
| | *Tcrg-C1* | 0.364 | 1.228 | 0.324 | −0.081 | 0.397 | |
| | *Trbc1* | 2.629 | 4.603 | 0.768 | 0.311 | 0.186 | −0.475 |
| | *Trdc* | 0.403 | 0.981 | 0.045 | −0.018 | 0.657 | −0.519 |
| | *Cd3g* | 2.773 | 8.154 | 1.616 | 0.195 | 0.466 | −0.583 |
| | *Cd3e* | 0.433 | 1.428 | 0.056 | | 0.582 | −0.462 |
| | *Cd3d* | 0.465 | 2.918 | 0.103 | −0.253 | 1.100 | −0.719 |

Average expression is Log2 scaled, expression change is calculated based on logged expression.
*Only log2FC > 0.01 and <−0.01 are shown here.

performed pathway analysis using all significantly upregulated markers called from each of the clusters. Different biological "hallmark" pathways from MSigDB were over-represented (Fig. 3a). Gene Set Enrichment Analysis (GSEA) analysis using expression matrix and mouse-ortholog hallmark gene sets as the *priori* reference further confirmed the hallmark pathways (Supplementary Fig. 1). Next, we assigned cluster identities by their over-represented hallmark pathways and expression profiles of selected genes-of-interest (GOI), inferring some of the important anti-tumoral immune functions (Table 1). The selected GOI are markers canonically known to be involved in different proposed ILC2 subpopulations[22] and mechanisms of immune functions such as Th1, Th2, T cell recruitment, antigen-presentation pathways (APP). The expression levels of GOI vary across each of the clusters (Fig. 3b). Most of the clusters maintained the key type 2 innate lymphoid markers. Within these clusters, C0 and C1 express the conventional ILC2 markers *Gata3, Il1rl1, Il5 and Il13*. We also found the previously identified inflammatory GATA3+ ST2lo IL25Rhi KLRG1+ ILC2 subtype C2, a subset of ILC2s that expands upon IL25 activation[49], displayed heightened IL25 receptor gene *Il17rb* expression as well as *Klrg1*. C3 has many cell-cycle enriched hallmark pathways, such as E2F pathway targets that are involved in cell proliferation and DNA damage repair; G2/M checkpoint pathway that can transit cells into mitosis; and mitotic spindle hallmark pathway which is represented by genes important for mitotic spindle assembly (Fig. 3a). These cell cycle-related pathways suggest that cells in C3

are undergoing proliferation. The C3 cells, instead of forming a discrete cluster based on distance matrix, some cells "blended" into other clusters on the tSNE plot (Fig. 2a) suggesting that cells from multiple clusters are undergoing proliferation. C6 is enriched for antigen presentation pathways (APP) and over-represented by class II histocompatibility antigen genes such as *H2-DMa, H2-Ab*, etc. *Cd74* plays an important role in MHC class II presentation and MHC class I cross-presentation, is also significantly upregulated in C3. The enriched APP-specific markers are consistent with previous and ongoing discovery of MHC class II antigen presentation equipped by subsets of ILC2s[22,50]. C7 is identified as the IL25R+ memory ILC2s for their small cluster size and heightened expression of *Il17rb* which encodes for the receptor for IL25. ILC2s expressing high levels of IL17RB or IL25R can respond to secondary and IL25 stimulation rapidly thus go through enhanced proliferation[70]. The aforementioned clusters all possess different GOI expression patterns yet retain strong type 2 ILC identity (*Gata3+, Il1rl1+, Il5+, Il13+*). These clusters are also enriched for T cell recruitment markers (*Cxcl10, Ccr2, Ltb*), an exciting feature that propounds their ability to facilitate other immune cells attraction/migration. Even though these ILC2s with T cell recruitment signatures do not show significant expression of *Ifng*, their receptor *Ifnγr1* is up-regulated in majority of the cells, suggesting that cells are ready to accept and process the IFN-mediated signals via paracrine interactions in the tumour-modified microenvironment. This observation is well in line with our preliminary data and with the

recent literature that demonstrates IFN type I/II's abilities to repress conventional type 2 immunity provided by ILC2 cells, in order to promote inflammatory responses[71,72]. On the other hand, some outlier clusters express intriguing, unexpected markers such as the interferon γ receptor gene *Ifnγr1* and other Th1-related genes in C4 and T cell-related markers in C5. In C4, cells downregulate *Gata3* and upregulate *Tbx21* and *Rorc* suggesting possible ILC2 identity shifts away from type 2 towards a type 1 and type 17 immunity influenced by the biological settings. Importantly, this cluster expresses higher levels of Th1-related markers such as IFN-induced genes (e.g. *Ifit1, Ifit3, Ifitm1, Gzmb* and *Gzmc*, Fig. 3b), strongly suggesting a genotype switch to pro-inflammatory type 1 immunity and acquired IFNγ-dependent characteristics. Given the pro-inflammatory potential and importance of these type 1 genes, we confirmed the expression of *Gzmb*$^+$, *Gzmc*$^+$ and *Ifitm1*$^+$ in a fresh ILC2 population by intracellular and surface staining followed by flow cytometry (Fig. 3c). C5 expresses significantly higher level of CD3 subunits (*Cd3δ, Cd3ε, Cd3γ*) and TCR-related components, such as genes encoding for constant fragments of the TCRβ/δ/γ (*Trdc, Trbc1, Tcrg-C1, Tcrg-C2*) and lymphoid specific factor *Tcf7*. These markers are typically expressed by T cells and convention- ally absent from the family of innate immune cells that lack antigen-specific receptors. Whether these TCR$^+$ cells are a contamination of T cells escaped from rigorous enrichment and sorting process or a true detection of new ILC2 subtype with potential adaptive immunity is up to debate and requires additional functional proof. Note the high expression of *Ifnγr1* (Fig. 3b) suggests that Th1 differentiation program may be involved in this cluster, especially in combination with significant down-regulation of the expression of conventional Th2-related characteristics. These unconventional ILC2 markers steer the cells away from the conventional type 2 immunity, forming the outlier clusters in the ILC2s (Fig. 2a). This observed heterogeneity defines ILC2 functional plasticity influenced by growing tumour microenvironment.

**Tumour presence induces ILC2 shift towards heightened pro-inflammatory immunity.** To compare how a growing tumour naturally affects the tumour-associated ILC2 (ta-ILC2) tran- scriptome, we conducted a comparative analysis of naïve-ILC2s without any stimulation, ta-non-activated-ILC2s (freshly isolated cells from the mouse lungs) and ta-activated-ILC2s (isolated cells from lungs and ex vivo stimulation with IL-33 and TSLP). At whole population level, ILC2s display differential expressions profiles influenced by the presence of tumour and activation status of the cells. Figure 4a is a heatmap showing the top 10 significantly upregulated genes in the respective treatment samples. Interestingly, while naïve-ILC2s and ta-non-activated ILC2s share similar top- upregulated markers and expression patterns, ta-activated ILC2s upregulates many Th1-related effector genes and IFN-induced markers, such as *Gzmc, Ifitm1, Ifitm2* and *Ifitm3*, along with the T cell recruitment signature gene *Cxcl10*. This strongly implies that ILC2s as a whole population, acquires an elevated reactive potential of the ta-ILC2 population after ex vivo stimulation, enabling them to gain associated effector functions.

Within the whole ILC2s, we observed similar heterogeneity in all samples with conventional ILC2s (C0,1) being the majority of the overall ILC2 populations. The conventional ILC2s showed little cluster size change across different conditions regardless of tumour development nor ILC2 activation (Fig. 4b, c). The stable expression of *Gata3* in the total ILC2s across different conditions remained mostly unchanged (log2FC < 0.01, Table 1) suggesting that ILC2 largely maintained their type 2 identity during tumour develop- ment and activation, at least at the transcriptional factor level, in the lungs. Downstream of *Gata3*, signaling products *Il5* and *Il13* showed enhanced expression in the ta-ILC2s–with and without IL33/TSLP stimulation relative to naïve, unstimulated ILC2s–which suggests that ILC2s may respond to distant neoplasm in a similar way as other viral and parasitic infections, and release type 2 cytokines. Despite of the stable maintenance of type 2 identity observed in the majority of the total ILC2s across different conditions, other clusters seem to have immunological plasticity that responds to the presence of tumour growth and IL33/TSLP stimulation (Fig. 4c, Table 2). In the presence of tumours, frequency of C2 *Klrg1*+ inflammatory cluster is drastically reduced from 19% in naïve-ILC2s to nearly 0 in ta-ILC2s in the mouse lungs. The average expression of *Klrg1* in total ILC2s synchronizes with this change by expressing the highest level in the naïve-ILC2 sample and lowest in the ta-activated ILC2s, resulting in a positive 0.35 log2-fold change (log2FC) in total naïve ILC2s compared to ta-ILC2s. Inflammatory ILC2s are known to be highly plastic towards a ILC3-like identity when cultured under Th17 conditions or exposed to infections. This is largely due to their basal level background RORγT expression (as observed in our naïve ILC2s, Table 1), and IL13 and IL17A productions[73,74]. In the case of emerging tumours, these inflammatory ILC2s may be going through similar genotypic changes and, based on mounting evidence[50,68,75,76], it is also highly likely that the plasticity of inflammatory ILC2s may extend to a Th1 shift, since these cells respond to IL-1 and IL-12 stimulation with cell activation, heightened T-bet expression, increased Th1 molecule production and lowered expression of GATA3. All this points to the attainment of ILC1-like phenotype and the disappearance of certain ILC2 subtypes like C2 KLRG1+ inflammatory ILC2s. This transcriptional reprogramming is usually triggered by certain immunological challenges such as infections and in this case, tumour development. The Th1-like cluster C4 also shrinks in cluster size in the tumour-associated samples. However, this is contradictory to the overall increased expression of Th1-related genes in the tumour-associated samples, especially the ta-activated ILC2s (Fig. 4a, d, Table 1). The decreased cell frequency of C4 rejects the idea that the pro-inflammatory type 1 shift in ta-ILC2s is contributed to by a discrete single sub-population of ILC2s committing to ILC1 but suggests a more generalized acquisition of Th1-like characteristics across the total ta-ILC2s, while retaining key type 2 signatures. This C4 frequency change may also be an artifact resulting from low cell numbers in this cluster (Table 2). Another cluster that disappears during tumour development is C7 IL25R+ memory ILC2s, dropping from 2% in naïve-ILC2s to close 0% in ta-acticated-ILC2s and completely missing in the ta-non-activated-ILC2 sample. In terms of expression of the memory ILC2 marker *Il17rb*, even though it seems that naïve ILC2s express slightly higher amount of *Il17rb* than ta-ILC2s (Fig. 4d), there is no significant expression change (log2FC < 0.01) in the overall ILC2 populations regardless of tumour exposure or ex vivo stimulation likely due to the small cluster size of the memory ILC2 subtype (Table 1). Like adaptive memory T cells, memory ILC2s can proliferate rapidly upon secondary stimulation and release huge amount of stored alarmin cytokines, facilitating fast and enhanced innate immune responses[70]. These memory ILC2s expand in the lungs followed by a contraction phase before migrating to lymph nodes for long-term residency[70]. This may explain the absence of memory ILC2s in our ta-ILC2 samples obtained from lung tissues as memory ILC2s may have already migrated out of lungs since the initial tumour exposure. Further functional and lineage tracing experiments are required to confirm this hypothesis. Additionally, under the development of tumours, ILC2s are proliferating, as reflected by the expansion of proliferating C3 by more than 5 fold in the ta-ILC2s comparing to naïve-ILC2s, and it is further increased upon ex vivo activation, indicating the active

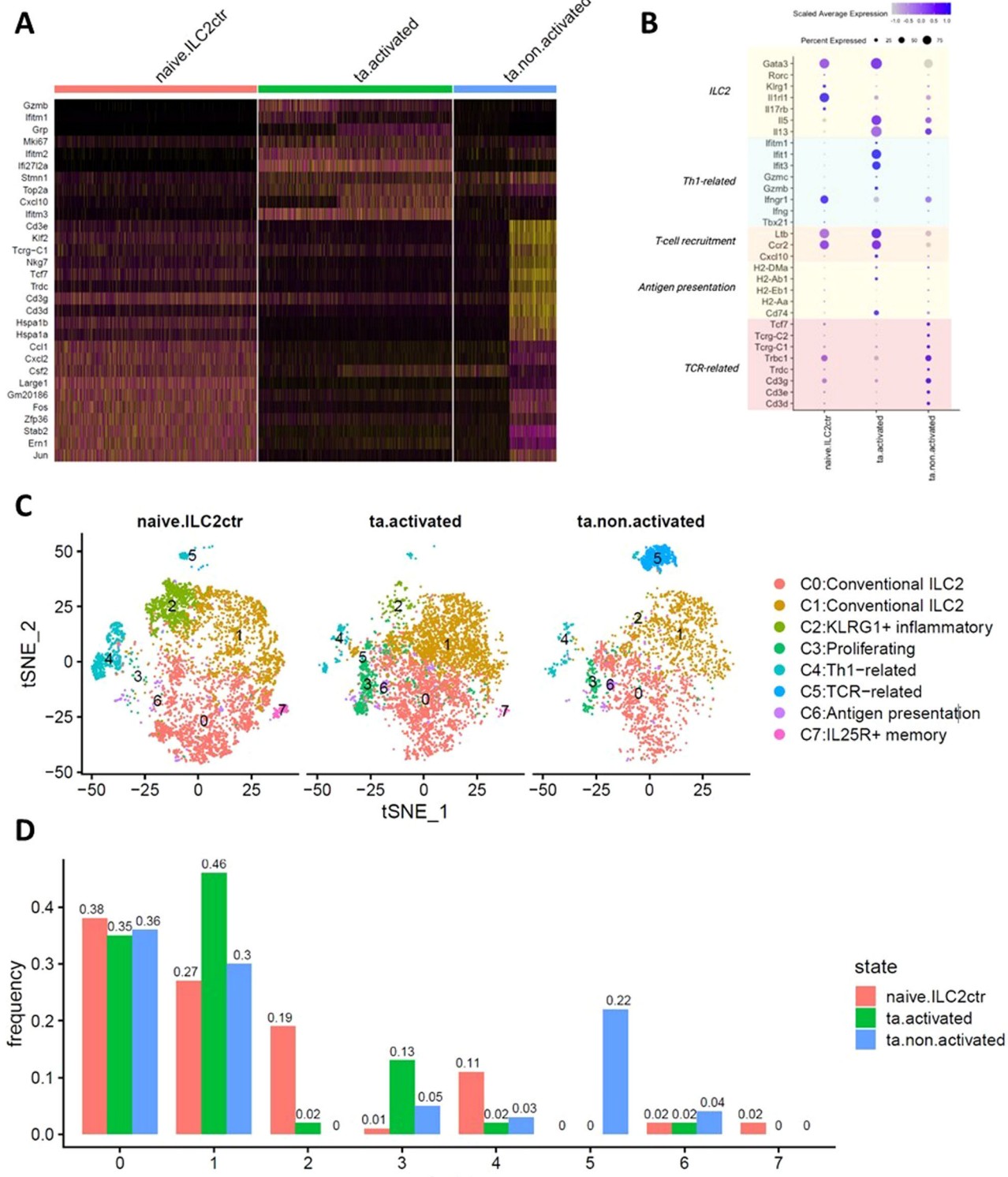

**Fig. 4 Comparative analysis reveals overall ILC2 and their subtypes shift towards heightened pro- inflammatory immunity. A** Heat map showing the top differentially expressed genes in naïve, ta-activated and ta-non-activated ILC2 samples, ranked by log-transformed fold change in expression levels in each sample. **B** t-distributed stochastic neighbour embedding visualization of distributions of clusters in the naïve and ta-ILC2s. Cluster sizes vary across different treatments. **C** Dotplot shows marker-of-interest expression per sample type. Color intensity indicates log-scaled mean gene expression level. Dot size indicates the fraction of cells in the cluster for each gene. **D** Cluster frequencies as the percentage of cells per cluster relative to the total number of cells, within naïve, ta-activated and ta-non-activated ILC2s. ILC2 subtypes react to different biological settings such as presence of tumour and ex vivo stimulation, and change subtype frequencies to exert different potential immune functions.

**Table 2 Cluster frequency change in the presence of tumours and ILC2 activation.**

| ILC2 Subtype | Cluster | Cell frequency | | |
|---|---|---|---|---|
| | | % in activated ta-ILC2s (cell number out of total 5840 cells) | % in non-activated ta-ILC2s (cell number out of total 3098 cells) | % in naïve-ILC2s (cell number out of total 6096 cells) |
| Conventional ILC2 | 0 | 35 (2029) | 36 (1117) | 38 (2321) |
| conventional ILC2 | 1 | 46 (2679) | 3 (923) | 27 (1616) |
| KLRG1+ inflammatory | 2 | 2 (117) | 0 (10) | 19 (1140) |
| Proliferation | 3 | 13 (742) | 5 (169) | 1 (36) |
| Th1-related | 4 | 2 (132) | 3 (92) | 11 (674) |
| TCR-related | 5 | 0 (1) | 22 (670) | 0 (13) |
| Antigen presentation | 6 | 2 (126) | 4 (117) | 2 (151) |
| IL25R+ memory | 7 | 0 (14) | 0 (0) | 2 (145) |

involvement and expansion of ILC2 in the face of tumour development. For the C6 antigen presentation subtype, cluster size is slightly increased in the ta-non-activated ILC2. Some of the antigen presentation-related GOI such as *H2-Dma, H2-Ab1*, and *Cd74* display increased expression levels in the ta-ILC2s, especially when ex vivo stimulated, whereas the other markers such as *H2-Eb1* and *H2-Aa* were unchanged, hinting toward additional antigen presentation mechanisms such as cross-presentation through MHC class I, in which CD74 plays a crucial role, or that CD74 may function as signaling transduction for cytokine production[77]. Furthermore, the most striking change is the C5 TCR-related cluster, dropping from 22% in the ta-non-activated-ILC2 sample to almost and complete absence (0%) in the other 2 samples (Fig. 4c), making the ta-non-activated total ILC2s express significantly higher TCR-related markers (Table 1). This unexpected cluster size and expression change is possibly due to a T cell population contamination in the ta-non-activated-ILC2 sample. For each sample type (i.e. naive-ILC2, ta-activated-ILC2, ta-non-activated-ILC2), cluster frequencies were calculated as the percentage of cells per cluster relative to the total number of cells that were sequenced and that passed the quality filters in that specific sample. It is important to note that the total number of sequenced cells vary between samples, and that this can lead to biased representations of certain rare subpopulations. In other words, the relative nature of cell abundance per cluster (i.e. cluster frequency) may not reflect the true distribution and should be interpreted with caution. Despite of this, based on the transcriptional profiles, ILC2 is a heterogeneous population wired with functional plasticity promoted by growing tumour environment, and this plasticity can be further inflated by IL33/TSLP stimulation.

Next, we examined the immunological plasticity within each of the clusters identified and observed that the immunological shift towards pro-inflammatory immunity is not only occurring at the whole population level, but individually the clusters acquire different characteristics that synergistically favours type 1 immunity. T cell recruitment markers are upregulated in the conventional, proliferating, Th1-related and memory ILC2s (C0,1,3,4,5,7), but less so in the inflammatory and TCR-related and antigen presentation subtypes (C2,5,6) when activated ex vivo (Fig. 5a). Type 2 identity is maintained in most of the individual clusters after activation as *Gata3* expression is retained and upregulated. Downstream *Il5* and *Il13* are also upregulated in most of the clusters, especially in the inflammatory C2 and memory C7 as a result of activation by potent IL33/TSLP. In the Th1-related C4 subtype, *Gata3, Il5* and *Il13* are expressed at much lower levels than the other clusters, but express increased amount of Th1-related, T cell recruitment and antigen presentation GOIs when activated. This is particularly interesting because the cells in this cluster still resemble traces of type 2 identity, yet upregulate genes that can potentiate unconventional type 1 immunity either by

directly functioning as type 1 cells, or indirectly by recruiting other immune cells to create a pro-inflammatory type 1 immune environment upon ex vivo stimulation, or both. It is important to note that in the TCR-related C5, it appears that ta-activated ILC2s greatly upregulated expression of type 2 markers, however, this result is only represented by one cell which demolishes any statistical significance.

Correlation analysis between activated and non-activated samples revealed genes that are consistently upregulated (*Ifi27l2a, Gzmb, Ifit1, Ifitm3, etc.*) and downregulated (*Il-9*) within many clusters (Fig. 5b). C5 and C7 are missing from this analysis due to the absence of these clusters in either the activated or non-activated sample.

Collectively, we identify a heterogeneous ILC2 population with dynamic immune subtypes that can be shifted towards a more pro-inflammatory genotype upon activation with main contribution to T cell recruitment power. This shift, a new developmental program, is observed in ILC2s globally, as well as within key immune clusters, suggesting that the ex vivo manipulations can modify transcriptional profile of ta-ILC2 cells, naturally primed by growing tumour, and better prepare them for cell-based immunotherapy.

## Discussion

Innate lymphoid cells are a group of lymphoid cells that have previously been demonstrated to lack functional antigen receptors such as those expressed on T and B cells, but are known to be largely tissue-resident cells that are deeply integrated into the fabric of tissue homeostasis[78]. More specifically, the role of ILC2s has now been established in the regulation of inflammation and immunity in tissues, including those of different cancers[20–22,78,79], establishing their potential role in cancer immunity and immunotherapy[20–22,66]. In this study, we aimed to investigate the interplay between tumours and ILC2s in the cancer environment and whether the effect that these two elements exert on each other is unidirectional or not.

To confirm that the presence of growing tumour induces the plasticity of ta-ILC2 cells by fundamentally changing their transcriptional profile, we performed scRNA-seq analysis on lung-derived ta-ILC2s at various time points during tumour development. To identify the potential power of the ta-ILC2 cells, which can be contributed to overall anti-tumour immune response, we conducted integrated analysis of ta-ILC2 cells isolated during week 2 and week 3 of tumour development as well as naïve-ILC2s from mice without tumour. Clusters' identities were distinguished by signature pathway genes of interests. In our single-cell transcriptome analysis, we found that main transcriptional programs, which had been enriched in ta-ILC2s, were responsible for T-cell recruitment, as well as for Th1- and Th2-related events. Thus, the immune system naturally generated a systemic pro-inflammatory

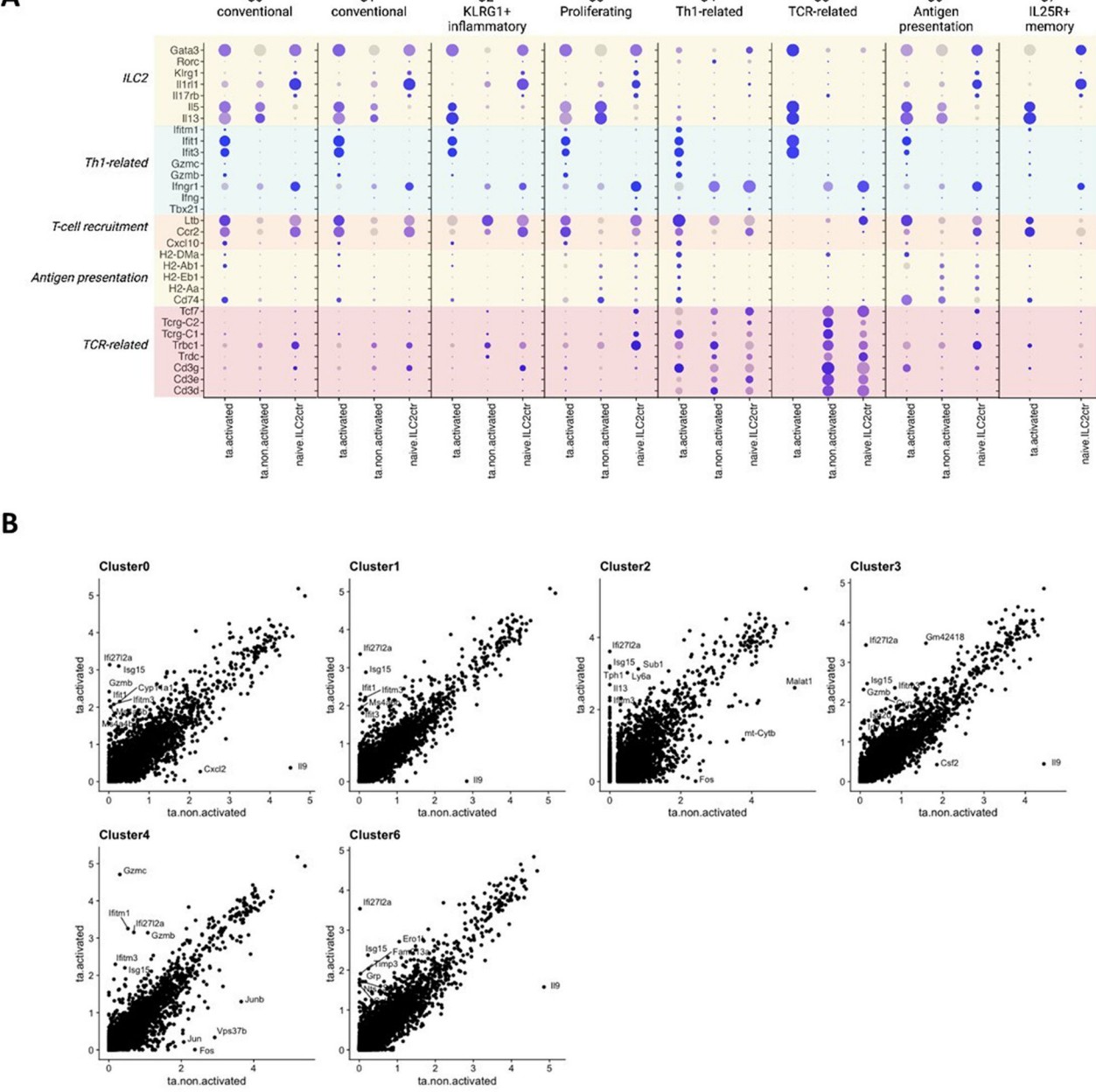

**Fig. 5 The immunological plasticity of ILC2 cells has been defined by a new developmental program responsible for heightened type 1 immune signature, regardless of the stability of Th2 signatures. A** Dotplot shows marker-of-interest expression within each of the clusters. The increased expression of Th1-related genes together with maintained expression of conventional Th2 identity in ta-ILC2 potentiates unconventional type 1 ILC2 immunity, especially upon ex vivo activation. **B** Correlation analysis showing outliers representing change of expression in response to IL-33 and TSLP activation within each cluster identified. The top 10 outliers with the most fold change per cluster are labelled in the plots. Genes that are consistently upregulated (*Ifi27l2a, Gzmb, Ifit1, Ifitm3*, etc) and downregulated (*Il-9*) within many clusters. C5 and C7 are missing from this analysis due to the absence of clusters in either of the activated or non-activated sample.

environment in response to neoplastic disease development at its early stages at distant organs such as the lungs.

The heterogeneous population of ILC2s presented different phases of cell development: early/proliferating, transitional, terminal and memory. The main portion of mature cells retained type identity and acquired heightened T cell recruitment and type 1 signatures (C0 and 1) expressed both Th2- and Th1-related features together with IFNγ receptor, whereas the population of developing ILC2s expressed Th2-related and proliferative markers. We confirmed the KLRG1+ inflammatory ILC2 subset (C2) has type 2 identity and increased strong expression of type 2

cytokines upon IL33/TSLP activation. A small subtype C4 demonstrated the simultaneous expression of high level of *Ifny* and *Granzyme b,c* genes together with expression of terminal ILC2 differentiation stage markers, which could possibly induce the local pro-inflammatory modifications of the microenvironment. An antigen presentation subset C6 expresses higher amount of antigen-presentation-related GOI, especially *Cd74*. Not only does CD74 play a crucial role in antigen presentation through MHC class II and cross-presentation through MHC class I by dendritic cells, it also functions as a receptor that can induce IL-8 production through NF-kB activation upon interaction with

migratory inhibitory factor (MIF)[80]. With antigen presentation abilities, this ILC2 subset may be able to instigate primary immune responses through interaction with antigen-specific T cells and may also facilitate/recruit adaptive immune cells. We also detected a small subset of ILC2s (C7) that differentiates into a memory subset and potentially migrates out of the lung tissue onset of tumour neoplasm at a distant location. Thus, the existence of clusters with different functional potentials, provide possibilities for subtle and synergistic regulation of immune response stimulated by growing tumour.

Due to transcriptional commonalities in ILC2s and ILC1s, their developmental programmes are responsible for low expression of Granzyme B[81] and therefore for the ILC2 plasticity towards NK- or ILC1-like phenotypes. Granzyme-expressing ILC2s could exert an important role in anti-cancer immune response, as granzymes have immunomodulatory functions that go beyond cytotoxicity. For example, granzymes can regulate cross-presentation[82] by easing DC antigen-uptake and CTL-priming, by leukocyte extravasation[83] and macrophage activation[84,85]. As demonstrated before, ILC2s can help tumour cells overcome the antigen-presentation deficiencies and, at the same time, aid in cytolytic T lymphocyte responses and immunotherapy[20–22]. It is possible that the presence of granzyme B in ta-ILC2s is not only a consequence of plasticity induction, but also a potential mechanism that facilitates the acquisition and presentation of antigen from dying tumour cells, supporting Th1-related responses via this potential novel pathway. Thus, granzymes, together with IFNs, act as mediators of pro-inflammatory environment during tumour progression.

Another interesting cluster C5 presented high level of TCR-related events with enriched expression of *Cd3-* and TCR-related structures, which are not conventional for the family of innate immune cells that lacks antigen-specific receptors. This cluster expands in response to a tumour, suggesting a potential role in cancer immune surveillance. The recent literature highlights the expression of TCR-related components on ILC2 cells, such as LAT, LCK, and ICOS and others[86–89], with elevated transcriptional factors, such as ITK and IRF4[90]. Alternatively expressed CD3-subunits and non-canonical TCR-components could allow signal transduction events that alter ta-ILC2 fate and function. Future studies will be essential to elucidate the biological implications of this novel population of TCR + CD3+ta-ILC2 cells.

To verify our scRNA sequencing data, we performed flow cytometry to detect IFITM1 molecule in a separate naïve ILC2 cell population and detected the marker at a level consistent with the scRNA sequencing data. We were especially interested in *Ifitm1* because IFITM1 expression has been often associated with the NK cells[91], and in our study, we repeatedly see *Ifitm1* being upregulated throughout many ILC2 subtypes when they are activated. Interestingly, IFITM1, HLA-A and other IFN-induced genes seem to be negatively regulated all together by the oncogenic factor miR-19 in human cancer cell lines[92]. Perhaps, the down-regulation of IL-33 that was linked to the low HLA expression levels in human prostate tumours[20–22] is a part of the same miR-19-mediated regulatory network, which leads to failed activation of ILC2s and poor tumour outcomes.

Overall, the immune system can modify transcriptional profiles of ILC2s in response to ongoing immunological challenges, in our case tumour development. At the initial stages, the heterogeneity of ta-ILC2 cells is enforced by subtypes that express Th2- and/or Th1-related characteristics, together with T-cell chemo-attractants and IFN-related features. These unique subsets of ta-ILC2 cells define their functional plasticity and further supported by the change of expression profiles upon ILC2 activation. The differential expression of genes in non-activated and activated ta-ILC2s have been depicted by our scRNA sequencing study and presented here, for the first time, as a new developmental program for ta-ILC2s. This program provides an explanatory background for the immunological plasticity of ILC2s.

We also conducted a correlation analysis to show the exact changes acquired by each cluster of ta-ILC2s during ex vivo stimulation. The ex vivo stimulation protocol used for the cell preparation for our other adoptive transfer studies up-regulated *Ifi27l2a, Gzmb, Ifit1, Ifit2, Ifit3, Ifitm1, Ifitm3* and other Th1-related genes and persistently down-regulated *Il-9* gene in the activated ta-ILC2 subtypes in comparison to the non-activated ta-ILC2s. It is worth mentioning that *Ifi27l2a* is also an IFN-induced gene with anti-viral ability[93,94]. *Il-9* gene is particularly interesting for its role in anti-inflammatory function through T-reg promotion, which is downregulated in many ILC2 subtypes observed in our study. ILC2s are known to produce IL-9 as an autocrine signal for activation and resolution of inflammation[95]. In response to ex vivo activation, ta-ILC2s can reprogram its transcriptome to reduce production of *Il-9* and acquire the possibilities to subsequently promote an inflammatory condition through impairment of Treg activation. Consequently, the reduced average expression of some conventional Th2-related cytokines together with the increased expression of Th1-related features further supports the idea that, in the presence of growing neoplasm, ta-ILC2s are able to exert unconventional type1 immunity upon ex vivo activation.

At the outset of this study, we sought to assess the hypothesis that ILC2s are highly plastic and they exist as multiple subsets of ILC2s in the context of tumour immunosurveillance. Having supported this initial hypothesis, we present single-cell analysis that demonstrates the heterogeneity of ILC2 equipped with a multitude of immune functions from antigen presentation to T cell recruitment and Th1 CTLs. This presents an exciting potential of a 'Swiss Army knife' tool for anti-tumour therapy. These discoveries may have translational applications for cell-based immunotherapy. Certainly, it is remarkable that lung ILC2s appear to act as a depot of innate lymphoid cells responsive to peripheral immunological insults and our data support the conclusion that these ILC2 subpopulations participate in anti-tumour immunity. Future technical innovations may allow the direct examination, by single-cell sequencing, of the scarce tumour infiltrating ILC2s and to compare them to the subpopulations ta-ILC2s elicited in the peripheral tissues by subcutaneous tumours. Our observations on the role of ILC2s in tumour immuno-surveillance may inspire a novel cell-based immunotherapy for cancer. In conclusion, identifying new ILC2 subpopulation through the use of scRNA sequencing analysis provides a deeper understanding of the complex and dynamic nature of haematopoiesis, shedding new light on their specific functions within the immune system.

## Materials and methods

**Cell line: murine lung cancer model**. The TC1 cell line is a murine lung tumour model derived from primary lung epithelial cells of C57BL/6 mice immortalized using the amphotropic retrovirus vector LXSN16 carrying human papillomavirus (HPV) genes E6/E7, and subsequently transformed with pVEJB plasmid expressing the activated human c-Ha-ras oncogene[96]. TC1 cells display high expression of TAP-1 and MHC-I[97]. The cell line was grown in Dulbecco's modified Eagle medium, supplemented with 10% heat-inactivated fetal bovine serum, 2 mM l-glutamine, 100 U/ml penicillin, 100 μg/ml streptomycin, and 10 mM HEPES, at 37 °C, with the air supplemented with 5% $CO_2$. The cells were tested by IDEXX RADIL and found to be free of any contaminating viruses or mycoplasma.

**Mice.** C57Bl/6 *Mus musculus* were purchased from the Jackson Laboratory and maintained in the Centre for Disease Modeling at the University of British Columbia. Female mice were used at 4–8 weeks of age. 5 mice per group. These experiments were ethically approved by the Animal Care Committee (ACC) at the University of British Columbia (UBC). We have compiled with all relevant ethical regulations for animal use. Animals were maintained and euthanized under humane conditions in accordance with the guidelines of the Canadian Council on Animal Care.

**Tumour establishment.** TC1 tumour cells [50 μl of $5 \times 10^5$ in HBSS (ThermoFisher Scientific)] or HBSS vehicle control were injected into wild type (WT) animals subcutaneously (s.c.) into the right flank. Tumour growth was monitored by measuring tumour dimensions using callipers. Tumour length and width measurements were obtained three times weekly. Tumour volumes were calculated according to the equation tumour volume=lengthxwidthxheightxπ/6 with the length (mm) being the longer axis of the tumour. Animals were weighed at the time of tumour measurement.

**ILC2 isolation.** Lungs from mice were harvested and cut into small pieces using a razor blade and digested for 30 minutes at 37 °C in a shaker platform (200 rpm) in 10 mL of digestion media per 5 pairs of lungs. The digestion medium employed contained RPMI 1640 (Gibco #11875-093) with 100 U P/S and 10% FBS, as well as 1 mL collagenase/hyaluronidase and 1.5 mL DNase I per 5 pairs of lungs (StemCell #07912 and #07900 respectively).

The digested pieces of lung tissue were placed onto a 70 μm cell strainer. Using the plunger end of a 3 mL syringe, the tissue was mashed through the strainer and rinsed with 5 mL RPMI 1640 to a total of 15 mL. Cells were centrifuged for 6 minutes at 1600–1700 rpm. The supernatant was carefully removed and the pellet was re-suspended in 20 mL ammonium chloride solution (StemCell #07800) and incubated at room temperature for 5 min to lyse the erythrocytes. After neutralization with 30 mL of FACS buffer cells were counted and washed (6 min, 1600–1700 rpm) in a total volume of 50 mL (full 50 mL Falcon Tube). FACS buffer was made of DPBS (Gibco #14190-136) with 2% FBS.

After counting cells, they were re-suspended in the appropriate volume of FACS buffer to obtain $1^* 10^8$ cells/mL, and then they were enriched for ILC2s using an EasySep™ Mouse ILC2 Enrichment Kit (StemCell #19842), which reduces sorting time and increases ILC2 recovery both from naïve and IL-33–treated lungs[70]. We did not exclude any ILC2 subtypes during our FACS enrichment, however, we limited our presentation of the results to the ILC2 markers that we used for purification. The FACS enrichment was conducted according to known protocols. Enrichment resulted in a population of 99% of pure Lin- ST2+ CD127+ Thy1.2+ CD45+ ILC2s.

Prior to sorting, the cells were stained with FITC-conjugated lineage marker mouse antibodies purchased from Thermo Fisher Scientific (CD3ε/γ #11-0032-80, CD4 #11-0042-81, CD8α #11-0081-81, CD19 #11-0193-81, TCRβ #11-5961-81, NK1.1 #11-5941-81, TER119 #11-5921-81, CD11c #11-0114-81, CD11b #11-0112-41 and Ly-6G/C #11-9668-80) and the ILC2 positive markers purchased from BioLegend: PE-conjugated CD127 (IL-7 receptor, #135009), PerCP-Cy5.5-conjugated ST2 (IL-33 receptor, #145312), BV605-conjugated Thy1.2 (#140317) and BV421-conjugated CD45 (#103133). Antibody concentrations used at 1:200.

**Flow cytometry.** BD FACS Aria II machine was used to phenotypically assess and sort the cells. Antibodies used to block Fc receptors: CD16/32 (564220, BD Pharmingen). Antibodies used to exclude nonviable cells from flow cytometry: Fixable Viability Dye eFluor 780 (65-0865-14, Affymetrix eBioscience).

**Intracellular staining for flow cytometry.** ILC2s derived from TC1 tumour-bearing mice (ta-ILC2s) were fixed and permeated using an intracellular fixation and permeabilization buffer set (eBioscience #88-8824-00) and then stained in PBS (2% FBS) with the following mouse antibodies: PE-conjugated granzyme B antibody (BioLegend #372207) and APC-conjugated granzyme C (BioLegend #150803). The IFITM1 mouse monoclonal antibody was purchased from Proteintech (#60074-1-Ig) and conjugated with a FITC antibody labeling kit (Mix-n-Stain #92295).

**Single cell RNA sequencing sample preparation.** Two- and three-weeks following tumour implantation, ILC2s from lungs are enriched and sorted. For the ta-non-activated-ILC2 and naïve-ILC2 samples, cells were immediately sent for sequencing. For ta-activated ILC2s, cells were cultured in StemSpan™ SFEM II medium (StemCell #09655) supplemented with rhIL-2, recombinant mouse rmIL-33 and TSLP (10 ng/mL each) for ILC2 activation (Invitrogen #14-8332-80 and #14-8498-90). Cells were cultured for 7 days at 37 °C prior to scRNA-seq. Samples are sequenced using 10x Genomics platform at Vancouver Prostate Cancer Centre.

**ScRNA-sequencing and data analysis.** ILC2s from mice with TC1 primary tumours were sorted based on Lin- ST2+ CD127+ Thy1.2 + CD45+ expressions. Sorted cells were sequenced using the 10x Genomics platform and the Cell Ranger pipeline was used to align reads and generate matrices. Seurat was used for downstream analysis. We followed the recommended settings of the standard workflow[69] to create Seurat project for each sequencing data. Five Seurat projects were created: week2 non-activated, week2 activated, week3 non-activated, week3 activated and naïve control ILC2s. The 5 datasets were then integrated to enable further comparative analysis. Integration followed the standard workflow[69]. After integration, cell-cycle scoring and regression was performed to eliminate overwhelming effect of cell cycle heterogeneity[98]. The regression was done based on the differences between the G2/M and S phase scores as described in Seurat, thereby differentiation between cycling and non-cycling cells is still enabled. Week2 non-activated and week3 non-activated are labeled as the ta-non-activated group and compared to week2- and week3- combined ta-activated group. Clusters were visualized by non-linear dimensional reduction technique t-distributed stochastic neighbour embedding (t-SNE). A population is defined by graph-based clustering driven by distance matrix, then partitioned by shared expression patterns. The 'granularity' of the clustering was set to 0.2 and returned 8 populations to reflect the primary sources of heterogeneity without having too many divisions. Pathway enrichment analysis was performed using clusterProfiler and GSEA, enricher function for over-representation test[99–101], compared against Hallmark gene set downloaded from Molecular Signature Database[102].

**Statistics and reproducibility.** Data were analyzed with R. Differential expression in single cell RNA-seq analysis is presented by logarithmic fold change, significance $p$ is determined by Wilcoxon Rank Sum test, $p \leq 0.05$ was considered significant.

**Reporting summary.** Further information on research design is available in the Nature Portfolio Reporting Summary linked to this article.

## Data availability

All sequencing data and processed data files are deposited and available at GEO (accession number GSE246502).

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

## Acknowledgements

We thank Andrew Johnson and Justin Wong for technical assistance in flow cytometry, as well as Tara Stach and Ryan Vander Werff for their help in RNA sequencing. We also thank Bettie Yim, Kayla Apperloo, Rhonda Hildebrandt, Brian Ryomoto and Tracy Welch for their assistance in tumour inoculation in mice, and Drs Giorgia Caspani and Eliana al Haddad for proofreading and editorial assistance and comments on the study. PLF and IS were supported by a MITACS Accelerate (IT12482) Cluster Fellowship; respectively; CWX was supported by a 4YF Scholarship; WAJ was supported by a Canadian Institutes of Health Research (CIHR) Operating Grant (PJT-148923) and an Industrial-Partnered Collaborative Research Grant from the CIHR (IPR-139079), as well as an NSERC Discovery Grant (AWD-02473) and by donations to the laboratory of WAJ through the Sullivan Urology Foundation at Vancouver General Hospital (https://www.urologyfoundation.ca). The funding sources had no role in the study design, data collection, analysis or interpretation of data, or in the writing of the paper.

## Author contributions

C.W.X. designed the research, performed the research, analyzed the data and wrote and edited the paper. I.S. designed the research, performed the research, analyzed the data and wrote the paper. P.L.F. performed the research, analyzed the data and wrote the paper. C.G.P. designed the research and analyzed the data. S.B. performed the research, analyzed the data and edited the paper. L.M., A.H., Y.Y.L., and S.L.B. performed the research and analyzed the data. C.C. performed the research, analyzed the data and edited the paper. W.A.J. conceived the project, designed the research, analyzed the data, and wrote and edited the paper.

## Competing interests

The authors declare no competing interests.
