## [Peer Review File · Communications Biology]

Reviewers' comments:

Reviewer #1 (Remarks to the Author):

The authors designed a transfer experiment of primed tILC2 to recipient-animals bearing primary tumors. Subsequently analysis revealed tILC2s featured with high expression level of Ifngr and low levels of Il5 and Il13. They also conducted scRNA-seq of tILC2 with/without stimulation, and figured out the pro-inflammatory phenotypes of tILC2. Thus, the authors claimed that tILC2 could reduce tumor growth via up regulating type I immune responses. However, the manuscript currently fails short in precisely characterizing the overall relevance of tILC2s in the context of tumor development. Specific comments :

1. What's the distribution of other immune cells except for eosinophils in tumors after the transfer of ILC2s? Did they observe any changes of infiltrating ILC2s and effector T cells?
2. The quality of Fig1b is poor. They should show different donors with/without tILC2 transfer and isolate the tumors for picture.
3. Did they observe tILC2 in tumor? If not, what's the mechanism that tILC2 contribute to the reduction of tumor growth? They observe more infiltration of eosinophils in tumors. It is well known that IL-5 and IL-13 secreted by ILC2s promote eosinophil accumulation (Jesse C. Nussbaum. Nature. 2013). However, the authors showed that tILC2s produced much lower levels of IL-5 and IL-13. It seems that tILC2s might not contribute to the accumulation of eosinophils in tumors. They should provide more evidence to support the anti-tumor function of tILC2 and find out the possible mechanisms.
4. For the scRNA-seq data, they observed several cluster express TCR-related genes. That might be contamination of T cells or immature T cells. They should reanalyze the data.
5. There are several typos throughout the text and figures. For example, "INFy " in line 362, "cxcl0" in line 506. For Fig 3B, the text in the heatmap should be deleted.

Reviewer #2 (Remarks to the Author):

This article entitled "adoptive transfer of type 2 innate lymphoid cells reduces tumor growth" has been written by Saranchova and colleagues. It aims at proving that ILC2 can exert an anti-tumor effect by slowing tumor growth of the pulmonary model TC-1. The role of ILC2 in cancer has received more and more attention lately but the paper fails to support its main claim. In its current form, this work will not be successful in convincing scientists in the community and it would require a lot of additional experiments to do so.

My main concern is the lack of data about ILC2 coming from naïve mice and from tumor bearing mice not activated ex vivo. It is very hard to draw any conclusion given that appropriate controls are missing. It is at the moment impossible to distinguish the impact of the tumor from the impact of the ex vivo activation.

The first figure show interesting data but raises many questions:

We are told that tumor growth experiments lasted for 35 days but we only see data up until the 31st day. Moreover, there is no error bars on the last point in tILC2 treatment when it is quite unlikely that it is so small that it cannot appear. It would be interesting to see all the individual growths.

It is also crucial to complete the experiment by transferring ILC2 from healthy mice's lungs (activated and not activated) and by transferring ILC2 from tumor bearing mice not activated ex vivo. Without these data, the rest of the paper is meaningless as we cannot say if the antitumor effect we observed is dependent on ILC2 in general, if the fact the ILC2 come from tumor bearing mice is relevant or not or if the activation is the main factor explaining the efficiency.

I do not understand the information brought by fig 1B.

Authors show that the transfer of tILC2 increased eosinophilic infiltration into the tumor tissue. In my opinion, using antibodies to block IL5, a likely candidate to link ILC2 and eosinophils would be of interest. Looking inside the tumor to see if transferred ILC2s got in the tumor or not would also bring support to their claim.

I have never seen RNAseq data graph in such ways and it is hard to understand as such, plus the figure 5 comes in during fig 1 and we are told about clusters not defined at this moment. There is a clear need for reorganization.

Comments on fig 2:

How come we still see lin+ cell in the gate against ST2 if the cells have been gated as lin- CD127- previously, this need to be corrected.

Since we do not know about the impact of nILC2 on the tumor, it is impossible to conclude anything about the differences between nILC2 and tILC2 as we do not know of these differences have any impact on the tumor growth.

RNA seq data need to be verified by QPCR, especially since there is a clear discrepancy between RNA seq data from fig 2b/c and fig 2D. For instance, IL5 and IL13 that are statistically more expressed by nILC2 in the RNAseq but no difference is seen in protein.

Moreover, we are told that ILC2 were from the lung of tumor bearing mice 2 and 3 weeks after tumor cells injections and then activated but we do not know if the tILC2 are ILC2 from 2 weeks or 3 and if they were activated or not. The nature of nILC2 also need to be clarified.

The statistics performed (unpaired T-tests) are not appropriate for these data and need to be redone properly.

It is hard to believe that some cytokines from fig 2D like IL-3 or IP-10 are not differentially expressed. In the text, it is said that "the RNA sequencing analysis confirmed the correct lineage identity by gata3 gene expression in both ILC2 groups", however, all ILC express Gata 3 as it promotes their expression of the IL-7Ralpha and without th2 as a positive control and ILC1 and ILC3 as negative controls, this claim cannot be made.

ILC are a plastic population that will be changed by ex vivo culture. To study the expression of CD4 on ILC2 it needs to be done on freshly isolated.

The fact that they find CD3 mRNA can mean 2 things : either their population of ILC2 contained T cells when they performed the RNA seq, or this CD3 expression need to be verified at the protein level and can be meaningless if they cannot find it.

In the figure 3

When doing tSNE, the number of clusters that appear is defined by the experimenter so instead of regrouping cluster 0 and 1, it would be more logical to redo the analysis asking for 9 clusters instead of 10.

What cells were used to graph this tSNE? Activated or not? From 2 or 3 weeks?

Fig3 b the color scale is missing, without it, it is impossible to interpret the data. moreover, since we are told cluster 0 and 1 are alike, they should show the data from all clusters for the genes in 3B.

An ELISA testing the secretion of Granzyme B would comfort this finding, that also need to be compared with unactivated ILC2 minimum.

Later, the authors claim that "subtypes with preserved Th2 characteristics appear to be involved to immune cell recruitment and may demonstrate antigen presentation features". None of these claims has been proved. Remove these affirmations or do the experiments needed to support it.

Furthermore, they compare tILC2 with itself and affirm that it has shifted toward a pro inflammatory type. This, again, is not supported by the provided data. The authors need to compare it to a pro-inflammatory immune cell type as a positive control and to ILC2 not exhibiting these features, while using other techniques than RNAseq. RNAseq data are not sufficient to conclude anything on the abilities of cells. Expression of molecules have to be checked at the protein level and functional assays have to be performed to support this claim.

Reviewer #3 (Remarks to the Author):

In the present study, Iryna Saranchova and colleagues investigated the role of ILC2s isolated from lungs of tumour-bearing mice in tumour growth . They found that they are capable of significantly reducing the growth of tumours. In addition, adoptively transferred ex vivo ILC2 cells display enhancement of the pre-inflammatory expression profile with the ability to expand and recruit T cells. Their results suggest that these cells are involved in immune monitoring and may be developed as a cell-based anti-tumor immunotherapy. Although ILC2 plays a different role in tumor development, this study is still an interesting topic. The following suggestions are put forward for reference.

Specific comments:

- 1.No specific surface marker of ILC2 has been found. How to isolate and purify ILC2. How to identify the purity of ILC2? What is the purity of ILC2 cells isolated in this study?
- 2.It is known that the number of ILC2 cells in peripheral blood is very small, and the number of cells immersed in tumor tissue should not be enough. How can you guarantee a certain number of ILC2 cells for testing?

Reviewer #1 (Remarks to the Author):

The authors designed a transfer experiment of primed tILC2 to recipient-animals bearing primary tumors. Subsequently analysis revealed tILC2s featured with high expression level of *Ifng* and low levels of *Il5* and *Il13*. They also conducted scRNA-seq of tILC2 with/without stimulation, and figured out the pro-inflammatory phenotypes of tILC2. Thus, the authors claimed that tILC2 could reduce tumor growth via up regulating type I immune responses. However, the manuscript currently fails short in precisely characterizing the overall relevance of tILC2s in the context of tumor development.

Firstly, we would like to thank you for taking the time to review our manuscript and provide your valuable comments. We would like to inform you that following the comments of the other referees, we have removed the data from the manuscript pertaining to adoptive transfer as it appears to be controversial in the review and requires substantial additional experimentation beyond the experiments we previously included. *In the present manuscript, we have refocused the paper on the single cell experiment and extended this analysis.* We have added substantial numbers of new experiments, including the FACS validation including the presence of a small percentage of CD3e+ is found in non-activated ILC2 population, and is further reduced to almost none in the activated ILC2s. We have now focused the study on demonstrating that subcutaneous tumours influence lung ta-ILC2 heterogeneity including the development of pro-inflammatory subsets that may support Th1-related anti-tumour responses. The existence of these new subtypes of tumour elicited ta-ILC2s challenges current paradigms of ILC2 biology. Overall, prompted by the referee’s comments we have also added several new datasets and provided a substantially expanded examination of the new ILC2 subsets we have discovered.

Specific comments :

1. What’s the distribution of other immune cells except for eosinophils in tumors after the transfer of ILC2s? Did they observe any changes of infiltrating ILC2s and effector T cells?

Referencing from Saranchova et al., 2018, showing immune infiltration of different immune cells in tumours of various stages and IL-33 levels. The expression of the ILC2 activator by the tumours

diminished the accumulation of immune suppressive cells and promoted elevated numbers of infiltrating tumour associated macrophages and neutrophils. In this paper we demonstrate that the ILC2 population has the potential to exert both conventional unconventional immune responses through the expression of different Th markers. A mechanism of action of ILC2s in enhancing CTL responses is detailed in Saranchova et al., 2018 paper.

2. The quality of Fig1b is poor. They should show different donors with/without tILC2 transfer and isolate the tumors for picture.

Figure1b has now been removed.

3. Did they observe tILC2 in tumor? If not, what's the mechanism that tILC2 contribute to the reduction of tumor growth?

We have removed these experiments from the current manuscript and have focused on single-cell analysis which remains novel, timely and important to immunologists and oncologists.

From our transcriptomic profiling data using the single cell sequencing, a subpopulation of ILC2s are equipped with T cell recruitment/attractant: Cxcl10, Ccr2 and Ltb, as well as pro-inflammatory and pro-survival signaling. Based on the presence of the large subtype (cluster 0+1), we propose that one of the mechanisms is through augmenting T cell migration and CTL differentiation at the tumour site, where ILC2s are expected to be present through IL33 signaling. Our paper, *Type 2 Innate Lymphocytes Actuate Immunity Against Tumours and Limit Cancer Metastasis*, Saranchova et al., (2018) doi: 10.1038/s41598-018-20608-6 details a mechanism of action of ILC2s during tumour development in augmenting Th1 responses.

They observe more infiltration of eosinophils in tumors. It is well known that IL-5 and IL-13 secreted by ILC2s promote eosinophil accumulation (Jesse C. Nussbaum. Nature. 2013). However, the authors showed that tILC2s produced much lower levels of IL-5 and IL-13. It seems that tILC2s might not contribute to the accumulation of eosinophils in tumors. They should provide more evidence to support the anti-tumor function of tILC2 and find out the possible mechanisms.

We published data on eosinophil infiltration in response to IL33 in our paper: *Type 2 Innate Lymphocytes Actuate Immunity Against Tumours and Limit Cancer Metastasis*, Saranchova et al., (2018) doi: 10.1038/s41598-018-20608-6. In our current study, we also show that ILC2 is a very heterogeneous population which has the potential to exert both Th1 and Th2 functions, therefore ILC2s can contribute to eosinophils (Th2 axis) and CTLs through Th1 signaling axis. One of the mechanisms of such anti-tumorigenicity we investigated is the role of ILC2 in mediating MHC-I antigen cross-priming for Th1 CTL responses. This study has been submitted for publication and is provided as a supplementary file to demonstrate the potential mechanism.

4. For the scRNA-seq data, they observed several cluster express TCR-related genes. That might be contamination of T cells or immature T cells. They should reanalyze the data.

The ILC2 populations were purified as lineage marker negative cells, with a purity exceeding 99.2%. Therefore, it is unlikely that there is contamination by T cells or immature T cells in the purified populations that we studied.

Specifically, the ILC2s acquired and used for sequencing went through vigorous enrichment (EasySep Mouse ILC2 Enrichment Kit) and FACS (Refer to new Figure 1 below) where we have included all of the purification regiments for each of the ILC2 populations that we perform scRNA sequencing analysis on.

Figure 1: Gating strategy for isolation of ILC2s from lungs. The gating of a standalone experiment to illustrate the getting strategy, since gating strategy is the same for all ILC2 isolations. In all cases this purification method yielded ILC2s with a purity of greater than 99%. ILC2 cells were sorted from Lungs of tumour bearing animals by FACS as Lin⁻ ST2⁺ CD127⁺ CD90.2⁺ cells. Sequential gating strategy was based on cell size [P1: small and non-granular cells, forward scatter (FSC) versus side scatter (SSC)], depletion of doublets (P2: SSC vs. Trigger Pulse Width), and selection of CD45⁺ cells. Cells with lineage-related markers were rigorously depleted during isolation process and the resultant cell numbers at each stage of purification is shown. The final gate (double positive for ST2 and Thy1.2) includes ILC2s isolated and sorted from lung tissue.

We have also conducted representative purification of populations of ILC2 at week 2 and 3 (but only show the representative purification scheme of week 1 (see above). These have now been included in the Appendix 1 Figure 1. We are including these here simply to show the referees that the ILC2 we examine are routinely over 99% pure. Please see below:

Appendix 1 Figure 1:

Shown below are 2 additional representative purification of populations of ILC2s at week 2 and 3 in addition to that shown in Figure 1. The ILC2 we examine in this study are routinely over 99% pure. Gating strategy: Gating strategy for ILC2s from lungs were sorted by FACS as Lin⁻ ST2⁺ CD127⁺ CD90.2⁺ cells. ILC2s isolated from untreated animals (0.1% of total). Sequential gating strategy was based on cell size [small and non-granular cells, forward scatter (FSC) versus side scatter (SSC)], depletion of doublets (FSC vs. Trigger Pulse Width), and selection of CD45⁺ cells. Cells with lineage-related markers were rigorously depleted during isolation process and the resultant cell numbers at each stage of purification is shown. ILC2s isolated from IL-33-treated animals (0.3% of total). The final gate (double positive for ST2 and Thy1.2) includes ILC2s isolated and sorted from lung tissue. Purified Lin⁻ ST2⁺ CD127⁺ CD90.2⁺ ILC2s express GATA3, IL5 and IL13 and this is stable during culturing.

Further, to address the referee's concern, we have included several new sets of confirmatory data and the identity of the sorted ILC2s isolated from naive mice ~8wks old with no tumours, in vivo IL33 activated and non-activated by flow cytometry using the lineage markers used for FACS sorting. All the lineage markers (negative selection) remained mostly negative (CD4, CD8 α , CD19, NK1.1, TER119, CD11c, CD11b and Ly-6G/C) and it should be noted that the percentage positive CD3e cells exceeds the sum of the very low frequency of CD4 positive cells added with the CD8 positive cells supporting the concept that this ILC2 CD3e positive subpopulation exists. Furthermore, the frequency of these cells in the validation FACS we have now included, corresponds well to the size and frequency of the cluster in the Single-cell sequencing paper. Finally, a gamma/delta TCR positive population of ILC2s has been previously described though the relationship with the ta-ILC2 population we have identified has yet to be explored (S. B. Shin *et al.*, Abortive gamma-delta TCR rearrangements suggest ILC2s are derived from T-cell precursors. *Blood Adv* **4**, 5362-5372 (2020)). Figures below have now been added to the appendix as Appendix 1 Figure 2 of the revised manuscript to elaborate on this point and to address the referee's comment.

Appendix 1 Figure 2: Lineage profiles. The identity of the sorted ILC2s is confirmed again from ILC2's isolated from naive mice ~8wks old with no tumours, in vivo IL33 activated and non-activated by flow cytometry using the lineage markers used for FACS sorting. All the lineage markers (negative selection) remained mostly negative (CD4, CD8 α , CD19, NK1.1, TER119, CD11c, CD11b and Ly-6G/C) and it should be noted that the percentage positive CD3e cells exceeds the sum of the very low frequency of CD4 positive cells added with the CD8 positive cells supporting the concept that this ILC2 CD3e positive subpopulation exists. Finally, a gamma/delta TCR positive population of ILC2s has been previously described though the relationship with the ta-ILC2 population we have identified has yet to be explored (S. B. Shin et al., Abortive gamma-delta TCR rearrangements suggest ILC2s are derived from T-cell precursors. *Blood Adv* 4, 5362-5372 (2020).).

Lineage markers confirmation

Activated

non-activated

CD3e

IL33-CD3e.fcs
live
27174

PBS-CD3e.fcs
live
6005

IL33-CD4.fcs
live
16779

PBS-CD4.fcs
live
7805

IL33-CD8A.fcs
live
20613

PBS-CD8A.fcs
live
6418

IL33-CD11C.fcs
live
18949

PBS-CD11C.fcs
live
6907

IL33-CD11D.fcs
live
17297

[Error: Unknown annotation type: PBS-CD11D.fcs]
live
6751

IL33-CD19.fcs
live
23666

PBS-CD19.fcs
live
7339

Overall, the higher level of CD3e expression may be contributed by the small ILC2 TCR-related subtype. In a separate ILC2s subtyping experiment, very low levels of CD3e were found, in-vivo IL33 treated lung resident ILC2s express even less CD3e at week 2+3 than PBS-treated control group (**Figure 8**). This finding confirms our single cell sequencing data that TCR-related component expression is very low and almost exclusively found in non-activated ILC2s, the presence of this marker is further reduced to almost absent in activated ILC2s. Again, supporting

data now exists in the literature findings (Shin et. al., 2020) that opens a new area for exploration: ILC2s exhibit non-functional TCR gene rearrangement similar to $\gamma\delta$ T cells and may have T cell development origins. Our new data suggests that as ILC2s matures by the activation of IL33, they may lose their T cell origin signatures.

We now state, “Another functionally important ta-ILC2 subtype (Cluster 5) expresses significantly higher level of CD3 subunits (*Cd3 δ* , *Cd3 ϵ* , *Cd3 γ*) and TCR-related components, such as genes encoding constant fragments of the TCR $\beta/\delta/\gamma$ (*Trdc*, *Trbc1*, *Tcr γ -C1*, *Tcr γ -C2*) and lymphoid specific factor Tcf7. *Cd3*- and *TCR*-related genes enrichment is an intriguing expression profile for the family of innate immune cells that lack antigen-specific receptors. The functions of such TCR-related components remain to be elucidated. Note the high expression of *Ifngr1* (Figure 2C) in this cluster suggests that Th1 differentiation program may be involved in this subtype, especially in combination with significant down-regulation of the expression of conventional Th2-related characteristics. “

Figure 8. Flow cytometry analysis of in-vivo activated and control ILC2s isolated from naive mice confirms identified cluster markers from RNA-seq data at week 2+3.

Cells are selected by singularity, cell sizes and viability first. Numbers in the gates are the percentage of gated sub-population from the total population. Overall, the total number of cells recovered from the lungs is much higher in the IL33 treated samples in comparison to the PBS treated naïve control.

Figure 8 is an independent experiment separate from the RNA-seq data (figure 2 and 3a), this experiment is a flow experiment to confirm the RNA data, this experiment was done on ILC2 cells from naïve mice no tumour, activated in vivo activated with IL33 and non-activated and culture for 1 week before flow cytometry. Appendix Figure 2 is another gating of ILC2 of another experiment, from this experiment the cells are isolated and cultured in plate for one week before the flow experiment, this experiment is here solely for the purpose of confirming the lineage markers. In new Figure 8, we also confirmed the markers by Flow cytometry analysis using the identified cluster markers from RNA-seq data. Cells are selected by singularity, cell sizes and viability first. Numbers in the gates are the percentage of gated sub-population from the total population. Overall, the total number of cells recovered from the lungs is much higher in the IL33 treated samples in comparison to the PBS treated naïve control. T cell recruitment cluster was tested as double positive of CCR2 and ICOS. With IL33 activation, percentage of cells with ICOS+ and CCR+ was increased, consistent with the RNA-seq data. For the antigen presentation pathway cluster, CD74 was used as marker. There is undetectable level of CD74+ cells in either of the activated and naïve populations. This may suggest that CD74 are either expressed at only RNA level, or the proteins are expressed at only RNA level, or the proteins are expressed intracellularly that the antibodies are unable to detect. For TCR-related cluster, CD3e was used as a detection marker. We observed this very small percentage of ILC2s that are CD3e high in the naïve population, and the percentage dropped further when ILC2s are activated with IL33. For the Th1 cluster, IFITM and IFNgR1 were detected through flow cytometry at a consistent level as found in the RNA-seq data. All RNA-seq data are from wk2+wk3 combined.

Again, this is an additional new figure that has been added as new Figure 8 and we believe this addresses the concern of the referee.

5. There are several typos throughout the text and figures. For example, “INF γ ” in line 362, “cxcl0” in line 506. For Fig 3B, the text in the heatmap should be deleted.

Thank you for your observation. Figure 3B is now Figure 2A. The typo's have been corrected and the text has been edited so it is now legible. We have also added a legend for the heatmap to demonstrate the colour change. We thank the referee for pointing this out and this comment has increased the quality of the paper.

Figure 2. ta-ILC2s is a heterogeneous population with pro-inflammatory phenotypes, as a response to a developing neoplastic disease at week 2 and 3. A) The top ten differentially expressed genes per cluster. Heat map showing the top ten differentially expressed genes per cluster, ranked by log-transformed fold change in expression levels in each cluster compared with all other clusters. Molecular Signature Database Hallmark gene sets are used as reference. Numbers in parenthesis indicate the number of identified genes involved in the pathway overrepresentation test.

Reviewer #2 (Remarks to the Author):

This article entitled “adoptive transfer of type 2 innate lymphoid cells reduces tumor growth” has been written by Saranova and colleagues. It aims at proving that ILC2 can exert an anti-tumor effect by slowing tumor growth of the pulmonary model TC-1. The role of ILC2 in cancer has received more and more attention lately but the paper fails to support its main claim. In its current form, this work will not be successful in convincing scientists in the community and it would require a lot of additional experiments to do so. My main concern is the lack of data about ILC2 coming from naïve mice and from tumor bearing mice not activated ex vivo. It is very hard to draw any conclusion given that appropriate controls are missing. It is at the moment impossible to distinguish the impact of the tumor from the impact of the ex vivo activation.

Firstly, we would like to thank you for taking the time to review our manuscript and provide your valuable comments. We would like to inform you that following the comments of the other referees, we have removed the data from the manuscript pertaining to adoptive transfer as it appears to be controversial in the review and requires substantial additional experimentation beyond the experiments we previously included. In the present manuscript, we have refocused the paper on the single cell experiment and extended this analysis. We have added substantial numbers of new experiments, including the FACS validation including the presence of a small percentage of CD3+ is found in non-activated ILC2 population, and is further reduced to almost none in the activated ILC2s. We have now focused the study on demonstrating that subcutaneous tumours influence lung ta-ILC2 heterogeneity including the development of pro-inflammatory subsets that may support Th1-related anti-tumour responses. The existence of these new subtypes of tumour elicited ta-ILC2s challenges current paradigms of ILC2 biology. Overall, prompted by the referee’s comments we have also added several new datasets and provided a substantially expanded examination of the new ILC2 subsets we have discovered.

The first figure show interesting data but raises many questions: We are told that tumor growth experiments lasted for 35 days but we only see data up until the 31st day. Moreover, there is no error bars on the last point in tILC2 treatment when it is quite unlikely that it is so small that it cannot appear. It would be interesting to see all the individual growths.

We agree this is an interesting point to address in future experimentation following the comments of the other referees, we have removed these experiments from the current manuscript and have focused on single-cell analysis which remains novel, timely and important to immunologists and oncologists.

It is also crucial to complete the experiment by transferring ILC2 from healthy mice’s lungs (activated and not activated) and by transferring ILC2 from tumor bearing mice not activated ex vivo. Without these data, the rest of the paper is meaningless as we cannot say if the antitumor effect we observed is dependent on ILC2 in general, if the fact the ILC2 come from tumor bearing mice is relevant or not or if the activation is the main factor explaining the efficiency.

Again, following the comments of the other referees, we have removed these experiments from the current manuscript and have focused on single-cell analysis.

I do not understand the information brought by fig 1B. Authors show that the transfer of tILC2 increased eosinophilic infiltration into the tumor tissue. In my opinion, using antibodies to block IL5, a likely candidate to link ILC2 and eosinophils would be of interest.

Thank you, we have removed Fig 1B from the current manuscript. We have also removed these experiments from the current manuscript and have focused on single-cell analysis which remains novel, timely and important to immunologists and oncologists.

Looking inside the tumor to see if transferred ILC2s got in the tumor or not would also bring support to their claim.

We have removed the adoptive transfer experiments from the current manuscript and have focused on single cell analysis.

I have never seen RNAseq data graph in such ways and it is hard to understand as such, plus the figure 5 comes in during fig 1.

To address this concern, we have included additional analysis and depictions summarizing the data in Figures 2, 4, 5, 6 and Tables 1 and 2 and Figure 5 has now been moved in the revised, Single Cell analysis manuscript to become Figure 7:

Figure 2C and D:

C) Expression of ta-ILC2 signature genes by cluster. Dot plot shows marker-of-interest expression per cluster. Color intensity indicates log-scaled mean gene expression level. Dot size indicates the fraction of cells in the cluster for each gene. D) Intracellular staining for granzymes B & C: a portion of ta-ILC2s are positive for both (10.4%), but mostly for granzyme B (60.9%). A fraction of ta-ILC2s is positive for intracellular IFITM1 (1.4%) confirmed by flow cytometry. Tumours were isolated from the lungs of animals after 2+3 weeks of tumour growth in-vivo.

MSigDB

Figure 4. Molecular Signature Database (MSigDB). Pathway over-representation analysis using up-regulated marker genes within each cluster, Molecular Signature Database Hallmark gene sets are used as reference. Numbers in parenthesis indicate the number of identified genes involved in the pathway overrepresentation test. Tumours were isolated from the lungs of animals after 2+3 weeks of tumour growth in-vivo.

KEGG

Figure 5. Kyoto Encyclopedia of Genes and Genomes (KEGG). Pathway over-representation analysis using up-regulated marker genes within each cluster, Kyoto Encyclopedia of Genes and Genomes gene sets are used as reference. Numbers in parenthesis indicate the number of identified genes involved in the pathway overrepresentation test. Tumours were isolated from the lungs of animals after 2+3 weeks of tumour growth in-vivo.

Figure 6. Comparative analysis reveals ta-ILC2 subtypes composition shifts towards heightened pro-inflammatory immunity. A) t-distributed stochastic neighbour embedding visualization of clustering of the ta-ILC2 subsets using Seurat. Cluster identities are assigned and labelled by important immune functions. B) Dot plot shows gene expression levels (colour intensity) and percentage expressed (dot size) in the *ex vivo* activated ta-ILC2 cells characterized by the enrichment

of T cell recruitment power. C-G) Expression of ta-ILC2 marker-of-interests for different immunological significance in the whole population. Tumours were isolated from the lungs of animals after 2+3 weeks of tumour growth in-vivo.

A new developmental program for tILC2 cells

Figure 7. The immunological plasticity of tILC2 cells has been defined by a new developmental program responsible for low IL-13 expression, regardless of the stability of Th2 signature transcriptional factor (**Gata 3**). The increased expression of Th1-related genes together with reduced average expression of conventional Th2 identity in tILC2 cells primed by growing tumour are able to exert unconventional *type1 ILC2 immunity* upon *ex vivo* activation. Tumours were isolated from the lungs of animals after 2+3 weeks of tumour growth in-vivo.

Table 1. Marker gene expression log fold changed unique to the subtypes identified. Only genes past 0.01 logFC threshold + minimum of 1% of cells in either of the populations are shown with logFC.

Sub-type	Cluster	Marker	Expression logFC	P-value
TCR-related	5	Cd3δ	1.687309	5.689119e-242
		Cd3ε	1.044359	2.64E-273
		Cd3γ	1.17396	4.196524e-276
		Trdc	1.16898	2.526142e-48
		Trbc1	0.9298762	3.777252e-65
		Terg-C1	0.8019613	4.379481e-62
		Terg-C2	1.019686	1.26E-106
		Tcf7	1.010792	7.122172e-155
T cell recruitment	0 and 1	Cxcl10	0.9039502	5.18E-46
		Ccr2	0.5712905	3.13E-238
		Ltb	0.729153552	2.08E-253
APP	4 and 7	Cd74	0.8249184	1.181281e-103
		H2-Aa		
		H2-Eb1		
		H2-Ab1		
		H2-Dma		
Th1-related	9	Ifny		
		Ifngr1	0.5688722	9.23E-28
		Gzmb	1.0420539	1.645999e-18
		Gzmc	2.128825	7.16E-23
		Ifitm1	1.960315	7.05E-35
		Ifit1		
		Ifit3		
Th2-related	3	IL-5	0.6352907	1.62E-167
		IL-13	0.4618801	1.959389e-124
		Il1r1l		

Figure 8. Flow cytometry analysis of in-vivo activated and control ILC2s isolated from naive mice confirms identified cluster markers from RNA-seq data at week 2+3. Cells are selected by singularity, cell sizes and viability first. Numbers in the gates are the percentage of gated sub-population from the total population. Overall, the total number of cells recovered from the lungs is much higher in the IL33 treated samples in comparison to the PBS treated naïve control.

Figure 8 is an independent experiment separate from the RNA-seq data (figure 2 and 3a), this experiment is a flow experiment to confirm the RNA data, this experiment was done on ILC2 cells from naive mice no tumour, activated in vivo activated with IL33 and non-activated and culture for 1 week before flow cytometry. Appendix Figure 2 is another gating of ILC2 of another experiment, from this experiment the cells are isolated and cultured in plate for one week before the flow experiment, this experiment is here solely for the purpose of confirming the lineage markers. In new Figure 8, we also confirmed the markers by Flow cytometry analysis using the identified cluster markers from RNA-seq data. Cells are selected by singularity, cell sizes and viability first. Numbers in the gates are the percentage of gated sub-population from the total population. Overall, the total number of cells recovered from the lungs is much higher in the IL33 treated samples in comparison to the PBS treated naïve control. T cell recruitment cluster was tested as double positive of CCR2 and ICOS. With IL33 activation, percentage of cells with ICOS+ and CCR+ was increased, consistent with the RNA-seq data. For the antigen presentation pathway cluster, CD74 was used as marker. There is undetectable level of CD74+ cells in either of the activated and naïve populations. This may suggest that CD74 are either expressed at only RNA

level, or the proteins are expressed at only RNA level, or the proteins are expressed intracellularly that the antibodies are unable to detect. For TCR-related cluster, CD3e was used as a detection marker. We observed this very small percentage of ILC2s that are CD3e high in the naïve population, and the percentage dropped further when ILC2s are activated with IL33. For the Th1 cluster, IFITM and IFNGR1 were detected through flow cytometry at a consistent level as found in the RNA-seq data.

Again, this is a brand new figure that has been added as new Figure 8.

Table 2. Average expression of marker genes in non-activated and activated ta-ILC2 cells. Tumours were isolated from the lungs of animals after 2+3 weeks of tumour growth in-vivo.

Sub-type	Marker	Average Expression (total)		logFC	P-value	Cluster	logFC		P-value
		non-activated	activated				(within subtype)		
TCR-related	Cd3δ	3.0789652	0.0904961	-1.319211	1.60E-254	5	-0.785719733	1	
	Cd3ε	1.53457157	0.04859117	-0.8825771	0		-0.395596932	1	
	Cd3γ	8.598055	1.678885	-1.27616	6.09E-148		-0.871919973	0.00017494	
	Trdc	1.051338	0.039052				0.614663766	1	
	Trbc1	5.0293473	0.7496037	-1.237249	7.93E-115		-0.305195753	1	
	Terg-C1	1.27193	0.304324				-0.849141561	1	
	Terg-C2	1.19551541	0.02011703				-0.749993459	1	
	Tcf7	1.53392479	0.05448842	-0.8767137	0				
T cell recruitment	Cxcl10	0.4360839	3.339602			0 and 1	1.338251386	5.03E-40	
	Ccr2	1.623493	5.496507	0.9067579	0		0.450790623	4.18E-27	
	Ltb	2.272692	9.097650	1.12669	0.00E+00		0.825257283	3.57E-58	
APP	Cd74	0.7640469	1.7845043	0.4564593	3.71E-137	4 and 7	0.880275106	2.18E-36	
	H2-Aa	0.10190659	0.09528422						
	H2-Eb1	0.1053115	0.1052846						
	H2-Ab1	0.1256624	0.3505915				0.262568966	6.43E-31	
	H2-Dma	0.1854303	0.1685956						
Th1-related	T-bet	0.000836998	0.000498243						
	Ifnγ	0.1528331	0.0297946			9			
	Ifngr1	2.5028366	0.6376112	-0.7603345	3.55E-67		-1.74483543	0.242141575	
	Gzmb	0.1201169	8.1546847				2.399183816	0.017523005	
	Gzmc	0.011186	3.185915				5.100079295	1.32E-05	
	Ifitm1	0.03558842	2.3931564				2.699916106	0.000121341	
	Ifit1	0.05404779	6.5955353	1.9749228	0.00E+00		0.930325112	0.036502991	
	Ifit3	0.01072107	3.78932674	1.555726	0		0.980898479	0.417499576	
Th2-related	Gata3	3.292622	8.333463	0.7767083	0				
	IL-5	6.590831	9.640108	0.3376895	4.42E-184	3	0.461930206	1.25E-28	
	IL-13	46.08107	33.56717	-0.3089666	7.39E-108		-0.301059879	1	
	Il1rl1	1.176691	1.010364						

We are told about clusters not defined at this moment. There is a clear need for reorganization.

We thank you for your comment and again we have now reorganized this data and included additional experimental details (See above) to address this concern.

“ILC2s in the context of tumours have previously been considered as a homogenous population and unlike other more studied immune cells (e.g. T cells), there is limited markers for this population in the existing literatures. We are the first to discover the high degree of heterogeneity in this previously less defined population, and the purpose of this paper is to explore the effective subcutaneous tumours on the heterogeneity of ILC2 subpopulations and the potential immune functions inferred by their markers.

RNAseq data is commonly represented as tSNE plots to show cell relatedness to infer clustering of sub-populations. In order to compare expression levels in both of expression intensity (average expression) and abundance (percentage), dot-plots are used as they allow visualization of direct side-by-side comparisons between populations. Some paper also utilizing this visualization methods:

Muto, Y., Wilson, P.C., Ledru, N. et al. Single cell transcriptional and chromatin accessibility profiling redefine cellular heterogeneity in the adult human kidney. *Nat Commun* 12, 2190 (2021). <https://doi.org/10.1038/s41467-021-22368-w>

Syage, A.R., Ekiz, H.A., Skinner, D.D., Stone, C., O'Connell, R.M., and Lane, T.E. (2020). Single-Cell RNA Sequencing Reveals the Diversity of the Immunological Landscape following Central Nervous System Infection by a Murine Coronavirus. *J Virol* 94.”

Further, we have also now added the following to the manuscript:

“The Gata3 transcriptional factor, a conventional coordinator of Th2-related events, was stably overexpressed in all the activated clusters (except Clusters 5 and 9) and in the whole activated population of the tILC2ta-ILC2 cells. However, downstream signaling products, which were expected to be over produced in response to IL-33/TSLP stimulation, did not follow the Th2-model-response. In particular, while IL-5 gene was overexpressed simultaneously with Gata3 transcriptional factor, IL-13 gene was down-regulated in the majority of clusters and in the tILC2ta-ILC2 population in total (Figure 7, Table 2). At the same time, the expression of Tbx21 (T-bet) transcriptional factor was low in all the clusters regardless of activation, meaning that tILC2ta-ILC2 cell did not differentiate into ILC1 subtype. Nevertheless, the increased expression of Th1-related markers in the whole population together with overall reduced average expression of IL-13 gene, a conventional Th2 marker, upon stimulation, confirms the immunological plasticity of tILC2ta-ILC2 cells that has been defined by a new developmental program responsible for up-regulation of Th1-related and down-regulation of some Th2-related events in spite of the expression of signature transcriptional factors. This type of immunological plasticity further supports the idea that tILC2ta-ILC2s primed by growing tumour are able to exert unconventional type 1 ILC2 immunity upon ex vivo stimulation.”

Figure 8. Flow cytometry analysis of in-vivo activated and control ILC2s isolated from naive mice confirms identified cluster markers from RNA-seq data at week 2+3. Cells are selected by singularity, cell sizes and viability first. Numbers in the gates are the percentage of gated sub-population from the total population. Overall, the total number of cells recovered from the lungs is much higher in the IL33 treated samples in comparison to the PBS treated naïve control.

Figure 8 is an independent experiment separate from the RNA-seq data (figure 2 and 3a), this experiment is a flow experiment to confirm the RNA data, this experiment was done on ILC2 cells from naive mice no tumour, activated in vivo activated with IL33 and non-activated and culture for 1 week before flow cytometry. Appendix Figure 2 is another gating of ILC2 of another experiment, from this experiment the cells are isolated and cultured in plate for one week before the flow experiment, this experiment is here solely for the purpose of confirming the lineage markers. In new Figure 8, we also confirmed the markers at week 2+3 by Flow cytometry analysis using the identified cluster markers from RNA-seq data. Cells are selected by singularity, cell sizes and viability first. Numbers in the gates are the percentage of gated sub-population from the total population. Overall, the total number of cells recovered from the lungs is much higher in the IL33 treated samples in comparison to the PBS treated naïve control. T cell recruitment cluster was tested as double positive of CCR2 and ICOS. With IL33 activation, percentage of cells with ICOS+ and CCR+ was increased, consistent with the RNA-seq data. For the antigen presentation pathway cluster, CD74 was used as marker. There is undetectable level of CD74+ cells in either of the activated and naïve populations. This may suggest that CD74 are either expressed at only RNA

level, or the proteins are expressed at only RNA level, or the proteins are expressed intracellularly that the antibodies are unable to detect. For TCR-related cluster, CD3e was used as a detection marker. We observed this very small percentage of ILC2s that are CD3e high in the naïve population, and the percentage dropped further when ILC2s are activated with IL33. For the Th1 cluster, IFITM and IFN γ R1 were detected through flow cytometry at a consistent level as found in the RNA-seq data.

Again, this is a brand new figure that has been added as new Figure 8.

Comments on fig 2:How come we still see lin+ cell in the gate against ST2 if the cells have been gated as lin- CD127- previously, this need to be corrected.

Thank you for pointing that out. This was a mistake and we have now corrected that figure.

Populations: Cytometer			
Populations	Events	% Total	% Parent
All Events	500,000	100.000%	###
P1	262,851	52.570%	52.570%
P2	230,304	46.061%	87.618%
CD45+	14,293	2.859%	6.206%
CD127+	368	0.074%	2.575%
ST2+	254	0.051%	69.022%
ILC2	252	0.050%	99.213%

Figure 1: Gating strategy for isolation of ILC2s from lungs. The gating of a standalone experiment to illustrate the getting strategy, since gating strategy is the same for all ILC2 isolations. In all cases this purification method yielded ILC2s with a purity of greater than 99%. ILC2 cells were sorted from Lungs of tumour bearing animals by FACS as Lin⁻ ST2⁺ CD127⁺ CD90.2⁺ cells. Sequential gating strategy was based on cell size [P1: small and non-granular cells, forward scatter (FSC) versus side scatter (SSC)], depletion of doublets (P2: SSC vs. Trigger Pulse Width), and selection of CD45⁺ cells. Cells with lineage-related markers were rigorously depleted during isolation process and the resultant cell numbers at each stage of purification is shown. The final gate (double positive for ST2 and Thy1.2) includes ILC2s isolated and sorted from lung tissue.

Since we do not know about the impact of nILC2 on the tumor, it is impossible to conclude anything about the differences between nILC2 and tILC2 as we do not know of these differences have any impact on the tumor growth. RNA seq data need to be verified by QPCR, especially since there is a clear discrepancy between RNA seq data from fig 2b/c and fig 2D. For instance, IL5 and IL13 that are statistically more expressed by nILC2 in the RNAseq but no difference is seen in protein.

Thank you for your comment. In fact, in our previously published paper (Saranchova I, Han J, Zaman R, Arora H, Huang H, Fenninger F, Choi KB, Munro L, Pfeifer CG, Welch I, Takei F, Jefferies WA. Type 2 Innate Lymphocytes Actuate Immunity Against Tumours and Limit Cancer Metastasis. *Sci Rep.* 2018 Feb 13;8(1):2924. doi: 10.1038/s41598-018-20608-6. PMID: 29440650; PMCID: PMC5811448.), we were able to demonstrate, using an animal lacking ILC2's (knockout), that tumours grow more rapidly in said mice.

In this revised paper, we explore the effect of subcutaneous tumours on the heterogeneity of ILC2s in the lungs of the tumour bearing mice. We discovered key markers including many Th1 genes for sub-populations with activation, we infer that the ILC2 is a plastic population that can shift away from conventional Th2 transcriptional signatures, reflecting potential Th1 immunity against cancer. Several of the populations were verified by FACS.

We now state, “Another important ta-ILC2 subtype (Cluster 5) expresses significantly higher level of CD3 subunits (*Cd3δ*, *Cd3ε*, *Cd3γ*) and TCR-related components, such as genes encoding constant fragments of the TCRβ/δ/γ (*Trdc*, *Trbc1*, *Tcrγ-C1*, *Tcrγ-C2*) and lymphoid specific factor *Tcf7*. *Cd3*- and *TCR*-related genes enrichment is an intriguing expression profile for the family of innate immune cells that lack antigen-specific receptors. The functions of such TCR-related

components remain to be elucidated. Note the high expression of *Ifngr1* (Figure 2C) in this cluster suggests that Th1 differentiation program may be involved in this subtype, especially in combination with significant down-regulation of the expression of conventional Th2-related characteristics.”

We now include a new set of experiments to assess marker validation on isolated naïve ILC2s ex vivo week 1, either activated or non-activated by flow cytometry to confirm the presence of markers found in single cell data, this includes CD3e from Figure 3B and Appendix 1 Figure 2:

Figure 3: Heat map represents the top up-regulated markers in clusters 0 and 1.

B) The detection of the presence of a population of CD3e+ ILC2s. The non-activated ILC2s have 2.43% CD3e+ cells while the activated ILC2s have only 0.14% of CD3e+ cells by flow cytometry analysis. Appendix Figure 2 provides a survey of other lineage cell surface markers.

Moreover, we are told that ILC2 were from the lung of tumor bearing mice 2 + 3 weeks after tumor cells injections and then activated but we do not know if the tILC2 are ILC2 from 2 weeks or 3 and if they were activated or not. The nature of nILC2 also need to be clarified.

Thank you for this comment and for clarity. Week 2 + 3 data were merged and treated as one dataset to capture the changes induced by activation. In the case of non-activated samples, ILC2 cell numbers are low, therefore we merged the two time points to obtain a more statistically robust dataset. Activation status is labelled where relevant. Combining week 2 + week 3 data points allows us to look for significant changes that are specifically regulated by activation status.

Figure 3D. The statistics performed (unpaired T-tests) are not appropriate for these data and need to be redone properly. It is hard to believe that some cytokines from fig 3D like IL-3 or IP-10 are not differentially expressed.

In the text, it is said that “the RNA sequencing analysis confirmed the correct lineage identity by gata3 gene expression in both ILC2 groups”, however, all ILC express Gata 3 as it promotes their expression of the IL-7Ralpha and without th2 as a positive control and ILC1 and ILC3 as negative controls, this claim cannot be made.

Thank you for your observation. We have now removed that figure from the manuscript altogether.

Further, we did not mean that GATA+ = ILC2s. We apologise for this oversight. We have also changed the phrase to “The RNA sequencing analysis confirmed Gata3 positive gene expression in both ILC2 groups”.

In the paper we now state the following:

“The Gata3 transcriptional factor, a conventional coordinator of Th2-related events, was stably overexpressed in all the activated clusters (except Clusters 5 and 9) and in the whole activated population of the tILC2ta-ILC2 cells. However, downstream signaling products, which were expected to be over produced in response to IL-33/TSLP stimulation, did not follow the Th2-model-response. In particular, while IL-5 gene was overexpressed simultaneously with Gata3 transcriptional factor, IL-13 gene was down-regulated in the majority of clusters and in the tILC2ta-ILC2 population in total (Figure 7, Table 2). At the same time, the expression of Tbx21 (T-bet) transcriptional factor was low in all the clusters regardless of activation, meaning that tILC2ta-ILC2 cell did not differentiate into ILC1 subtype. Nevertheless, the increased expression of Th1-related markers in the whole population together with overall reduced average expression of IL-13 gene, a conventional Th2 marker, upon stimulation, confirms the immunological plasticity of tILC2ta-ILC2 cells that has been defined by a new developmental program responsible for up-regulation of Th1-related and down-regulation of some Th2-related events in spite of the expression of signature transcriptional factors. This type of immunological plasticity further supports the idea that tILC2ta-ILC2s primed by growing tumour are able to exert unconventional type 1 ILC2 immunity upon ex vivo stimulation.”

Table 2. Average expression of marker genes in non-activated and activated ta-ILC2 cells

Tumours were isolated from the lungs of animals after 2+3 weeks of tumour growth in-vivo.

Sub-type	Marker	Average Expression (total)		logFC	P-value	Cluster	logFC	P-value
		non-activated	activated					
TCR-related	Cd3δ	3.0789652	0.0904961	-1.319211	1.60E-254	5	-0.785719733	1
	Cd3ε	1.53457157	0.04859117	-0.8825771	0		-0.395596932	1
	Cd3γ	8.598055	1.678885	-1.27616	6.09E-148		-0.871919973	0.00017494
	Trdc	1.051338	0.039052				0.614663766	1
	Trbc1	5.0293473	0.7496037	-1.237249	7.93E-115		-0.305195753	1
	Tcrg-C1	1.27193	0.304324				-0.849141561	1
	Tcrg-C2	1.19551541	0.02011703				-0.749993459	1
	Tcf7	1.53392479	0.05448842	-0.8767137	0			
T cell recruitment	Cxcl10	0.4360839	3.339602			0 and 1	1.338251386	5.03E-40
	Ccr2	1.623493	5.496507	0.9067579	0		0.450790623	4.18E-27
	Ltb	2.272692	9.097650	1.12669	0.00E+00		0.825257283	3.57E-58
APP	Cd74	0.7640469	1.7845043	0.4564593	3.71E-137	4 and 7	0.880275106	2.18E-36
	H2-Aa	0.10190659	0.09528422					
	H2-Eb1	0.1053115	0.1052846					
	H2-Ab1	0.1256624	0.3505915				0.262568966	6.43E-31
	H2-Dma	0.1854303	0.1685956					
Th1-related	T-bet	0.000836998	0.000498243					
	Ifny	0.1528331	0.0297946			9		
	Ifngr1	2.5028366	0.6376112	-0.7603345	3.55E-67		-1.74483543	0.242141575
	Gzmb	0.1201169	8.1546847				2.399183816	0.017523005
	Gzmc	0.011186	3.185915				5.100079295	1.32E-05
	Ifitm1	0.03558842	2.3931564				2.699916106	0.000121341
	Ifit1	0.05404779	6.5955353	1.9749228	0.00E+00		0.930325112	0.036502991
	Ifit3	0.01072107	3.78932674	1.555726	0		0.980898479	0.417499576
Th2-related	Gata3	3.292622	8.333463	0.7767083	0			
	IL-5	6.590831	9.640108	0.3376895	4.42E-184	3	0.461930206	1.25E-28
	IL-13	46.08107	33.56717	-0.3089666	7.39E-108		-0.301059879	1
	Il1rl1	1.176691	1.010364					

ILC are a plastic population that will be changed by ex vivo culture. To study the expression of CD4 on ILC2 it needs to be done on freshly isolated.

ILC2s were sorted from freshly obtained lung tissues using lineage markers, which include CD4-gating. In fact, the expression of TCR-related markers (CD3, CD4, CD8, TCRb, NK1.1) was rigorously depleted from the surface during isolation process through both commercial selection kit and FACS. This freshly isolated sample (non-activated) was sent for RNA-seq without ex vivo culturing **Appendix 1 Figure 2**. We compared ILC2 cells from naive mice no tumour, activated in vivo with IL33 and culture for 1 week before flow cytometry and found a decrease in the expression TCR-related genes including CD4 * (**Appendix 1 Figure 2**).

The fact that they find CD3 mRNA can mean 2 things : either their population of ILC2 contained T cells when they performed the RNA seq, or this CD3 expression need to be verified at the protein level and can be meaningless if they cannot find it.

Figure 8. Flow cytometry analysis of in-vivo activated and control ILC2s isolated from naive mice confirms identified cluster markers from RNA-seq data at week 2+3.

Cells are selected by singularity, cell sizes and viability first. Numbers in the gates are the percentage of gated sub-population from the total population. Overall, the total number of cells recovered from the lungs is much higher in the IL33 treated samples in comparison to the PBS treated naïve control.

Figure 8 is an independent experiment separate from the RNA-seq data (figure 2 and 3a), this experiment is a flow experiment to confirm the RNA data, this experiment was done on ILC2 cells from naïve mice no tumour, activated in vivo activated with IL33 and non-activated and culture for 1 week before flow cytometry. Appendix Figure 2 is another gating of ILC2 of another experiment, from this experiment the cells are isolated and cultured in plate for one week before the flow experiment, this experiment is here solely for the purpose of confirming the lineage markers. We agree with the referees and in new Figure 8, we confirmed the clusters by Flow cytometry analysis (Protein level) using the identified cluster markers from RNA-seq data including the reduce expression of CD3e at week 2+3. Cells are selected by singularity, cell sizes and viability first. Numbers in the gates are the percentage of gated sub-population from the total population. Overall, the total number of cells recovered from the lungs is much higher in the IL33 treated samples in comparison to the PBS treated naïve control. T cell recruitment cluster was tested as double positive of CCR2 and ICOS. With IL33 activation, percentage of cells with ICOS+ and CCR+ was increased, consistent with the RNA-seq data. For the antigen presentation pathway cluster, CD74 was used as marker. There is undetectable level of CD74+ cells in either of the activated and naïve populations. This may suggest that CD74 are either expressed at only RNA level, or the proteins are expressed at only RNA level, or the proteins are expressed intracellularly that the antibodies are unable to detect. For TCR-related cluster, CD3e was used as one of the detection marker. We observed this very small percentage of ILC2s that are CD3e high in the naïve population, and the percentage dropped further when ILC2s are activated with IL33. For the Th1 cluster, IFITM and IFNgR1 were detected through flow cytometry at a consistent level as found in the RNA-seq data.

Again, this is a new figure that has been added as new Figure 8.

We also now include a new set of experiments added to the appendix to assess other marker validation on isolated naïve ILC2s at week 1, either activated or non-activated by flow cytometry to confirm the presence of markers protein found in single cell data, this includes CD3e. These new data have been included in the paper as Figure 6 and Appendix 1 Figure 2:

Figure 3: Heat map represents the top up-regulated markers in clusters 0 and 1.

B) The detection of the presence of a population of CD3e+ ILC2s. The non-activated ILC2s have 2.43% CD3e+ cells while the activated ILC2s have only 0.14% of CD3e+ cells by flow cytometry analysis. Appendix Figure 2 provides a survey of other lineage cell surface markers.

We now state, “ Another important ta-ILC2 subtype (Cluster 5) expresses significantly higher level of CD3 subunits (*Cd3δ*, *Cd3ε*, *Cd3γ*) and TCR-related components, such as genes encoding constant fragments of the TCRβ/δ/γ (*Trdc*, *Trbc1*, *Tcrγ-C1*, *Tcrγ-C2*) and lymphoid specific factor *Tcf7*. *Cd3*- and *TCR*-related genes enrichment is an intriguing expression profile for the family of innate immune cells that lack antigen-specific receptors. The functions of such TCR-related components remain to be elucidated. Note the high expression of *Ifngr1* (**Figure 2C**) in this cluster suggests that Th1 differentiation program may be involved in this subtype, especially in combination with significant down-regulation of the expression of conventional Th2-related characteristics.”

In the figure 3 When doing tSNE, the number of clusters that appear is defined by the experimenter so instead of regrouping cluster 0 and 1, it would be more logical to redo the analysis asking for 9 clusters instead of 10.

A population is defined by graph-based clustering driven by distance matrix, then partitioned by shared expression patterns. The ‘dimensionality’ is arbitrarily defined by users. We adjusted parameters in the clustering processes, including principle component selection, dimension reduction, clustering resolution as reviewer suggested clustering labels can be arbitrary. However, no method that we attempted automatically combined the two clusters. Since we used K-mean clustering, each cell is forced into one cluster or another. There must be enough transcriptomic differences between these two clusters to keep them separate, but for our immunological subtyping the differences seem to be contained as we see signature immune markers expressed similarly in both clusters (Fig 2A).

What cells were used to graph this tSNE? Activated or not? From 2 or 3 weeks?

Activated and non-activated ILC2s are included and Week 2 + Week 3 data were merged and to capture the changes induced by activation. In the case of non-activated samples, ILC2 cell numbers are low, therefore we merged the two time points to obtain a more robust dataset. Activation status is labelled where relevant. Combining Week2 + Week3 datapoints allows us to look for significant changes that are specifically regulated by activation status.

Fig3 b the color scale is missing, without it, it is impossible to interpret the data. moreover, since we are told cluster 0 and 1 are alike, they should show the data from all clusters for the genes in 3B.

Thank you for your comment we didn’t realize the colour scale was missing until the referee pointed this out. We have now included this in the revised figure. We have replaced Figure 3b with a new Figure 2A showing all the data. We have also updated the writing in the text to make it clear and legible as well as adding a legend to illustrate the colours of the heatmap.

Figure 2. ta-ILC2s is a heterogeneous population with pro-inflammatory phenotypes, as a response to a developing neoplastic disease at week 2 and 3. A) The top ten differentially expressed genes per cluster. Heat map showing the top ten differentially expressed genes per cluster, ranked by log-transformed fold change in expression levels in each cluster compared with all other clusters. Molecular Signature Database Hallmark gene sets are used as reference. Numbers in parenthesis indicate the number of identified genes involved in the pathway overrepresentation test.

An ELISA testing the secretion of Granzyme B would comfort this finding, that also need to be compared with unactivated ILC2 minimum.

Thank you for this comment. We confirmed including expression of genes for the *Granzymes B & C*, and *IFITM1*, an IFN α/γ -induced transmembrane protein involved in the transduction of anti-proliferative signals [49] are significantly higher in taILC2s (significantly higher (**Figure 2C, Table 1**)). We also now confirm Intracellular staining for Granzymes B & C: a portion of ta-ILC2s are positive for both (10.4%), but mostly for granzyme B (60.9%). A fraction of ta-ILC2s is positive for intracellular IFITM1 (1.4%) confirmed by flow cytometry. Together we believe these data addresses the point of the referee comparing Granzyme B in nonactivated and active ILC2s.

Later, the authors claim that “subtypes with preserved Th2 characteristics appear to be involved to immune cell recruitment and may demonstrate antigen presentation features”. None of these claims has been proved. Remove these affirmations or do the experiments needed to support it.

Thank you for this comment. We have now removed this passage.

Furthermore, they compare tILC2 with itself and affirm that it has shifted toward a pro-inflammatory type. This, again, is not supported by the provided data. The authors need to compare it to a pro-inflammatory immune cell type as a positive control and to ILC2 not

exhibiting these features, while using other techniques than RNAseq. RNAseq data are not sufficient to conclude anything on the abilities of cells. Expression of molecules have to be checked at the protein level and functional assays have to be performed to support this claim.

In this study, we discovered the biomarkers that are differentially expressed in heterogeneous subpopulations of ILC2s. These markers are conventionally known to involve different mechanisms of immune functions such as Th1, T cell recruitment, APP. While further functional data is needed to confirm their immunological roles in anti-tumour immunity, the positive and dynamic expression of such markers open a vast spectrum of immune potentials that are previously unknown to the field. We are interpreting the functionality of the sub populations of ILC2s to the biomarker genes in proteins they express, and we think this is a reasonable approach. In light of the above, we are also limiting our interpretations.

We did a protein marker validation experiment on isolated naïve ILC2s using flow cytometry analysis, either activated or non-activated to confirm the presence of markers found in single cell data, this includes CD3e.

Figure 8. Flow cytometry analysis of in-vivo activated and control ILC2s isolated from naïve mice confirms identified cluster markers from RNA-seq data at week 2+3.

Cells are selected by singularity, cell sizes and viability first. Numbers in the gates are the percentage of gated sub-population from the total population. Overall, the total number of cells recovered from the lungs is much higher in the IL33 treated samples in comparison to the PBS treated naïve control.

Figure 8 is an independent experiment separate from the RNA-seq data (figure 2 and 3a), this experiment is a flow experiment to confirm the RNA data, this experiment was done on ILC2 cells from naïve mice no tumour, activated in vivo activated with IL33 and non-activated and culture for 1 week before flow cytometry. Appendix Figure 2 is another gating of ILC2 of another experiment, from this experiment the cells are isolated and cultured in plate for one week before the flow experiment, this experiment is here solely for the purpose of confirming the lineage markers. In new **Figure 8**, we also confirmed the clusters by Flow cytometry analysis using the identified cluster markers from RNA-seq data at week 2 +3. Cells are selected by singularity, cell sizes and viability first. Numbers in the gates are the percentage of gated sub-population from the total population. Overall, the total number of cells recovered from the lungs is much higher in the IL33 treated samples in comparison to the PBS treated naïve control. T cell recruitment cluster was tested as double positive of CCR2 and ICOS. With IL33 activation, percentage of cells with ICOS+ and CCR+ was increased, consistent with the RNA-seq data. For the antigen presentation pathway cluster, CD74 was used as marker. There is undetectable level of CD74+ cells in either of the activated and naïve populations. This may suggest that CD74 are either expressed at only RNA level, or the proteins are expressed at only RNA level, or the proteins are expressed intracellularly that the antibodies are unable to detect. For TCR-related cluster, CD3e was used as a detection marker. We observed this very small percentage of ILC2s that are CD3e high in the naïve population, and the percentage dropped further when ILC2s are activated with IL33. For the Th1 cluster, IFITM and IFNgR1 were detected through flow cytometry at a consistent level as found in the RNA-seq data.

Again, this is a new figure that has been added as new Figure 8.

We also added a new flow cytometry experiment to the appendix that validates the marker expression including CD3e as Figure 3B and Appendix 1 Figure 2.

Figure 3: Heat map represents the top up-regulated markers in clusters 0 and 1.

B) The detection of the presence of a population of CD3e+ ILC2s. The non-activated ILC2s have 2.43% CD3e+ cells while the activated ILC2s have only 0.14% of CD3e+ cells by flow cytometry analysis. Appendix Figure 2 provides a survey of other lineage cell surface markers.

We thank the referee for these insightful comments and by reanalysing our data, reorganizing the paper and including substantial new data we believe we have now addressed the comments and concerns.

Reviewer #3 (Remarks to the Author):

In the present study, Iryna Saranchova and colleagues investigated the role of ILC2s isolated from lungs of tumour-bearing mice in tumour growth . They found that they are capable of significantly reducing the growth of tumours. In addition, adoptively transferred ex vivo ILC2 cells display enhancement of the pre-inflammatory expression profile with the ability to expand and recruit T cells. Their results suggest that these cells are involved in immune monitoring and may be developed as a cell-based anti-tumor immunotherapy. Although ilc2 plays a different role in tumor development, this study is still an interesting topic. The following suggestions are put forward for reference.

Firstly, we would like to thank you for taking the time to review our manuscript and provide your valuable comments. We would also like to inform you that we have focused the current manuscript on single-cell analysis which, we believe, remains novel, timely and important to the immunologists and oncologists.

Specific comments:

1. No specific surface marker of ilc2 has been found. How to isolate and purify ilc2. How to identify the purity of ilc2? What is the purity of ilc2 cells isolated in this study?

The ILC2 populations were purified as lineage marker negative cells, with a purity exceeding 99.2%. Therefore, it is unlikely that there is contamination by T cells or immature T cells in the purified populations that we studied.

Specifically, the ILC2s acquired and used for sequencing went through vigorous enrichment (EasySep Mouse ILC2 Enrichment Kit) and FACS (Refer to new Figure 1) where we have included the purification regiments below for each of the ILC2 populations that we perform scRNA sequencing analysis on so the referee can both follow the purification workflow but also assess the overall purity.

Figure 1: Gating strategy for isolation of ILC2s from lungs. The gating of a standalone experiment to illustrate the getting strategy, since gating strategy is the same for all ILC2 isolations. In all cases this purification method yielded ILC2s with a purity of greater than 99%. ILC2 cells were sorted from Lungs of tumour bearing animals by FACS as Lin⁻ ST2⁺ CD127⁺ CD90.2⁺ cells. Sequential gating strategy was based on cell size [P1: small and non-granular cells, forward scatter (FSC) versus side scatter (SSC)], depletion of doublets (P2: SSC vs. Trigger Pulse Width), and selection of CD45⁺ cells. Cells with lineage-related markers were rigorously depleted during isolation process and the resultant cell numbers at each stage of purification is shown. The final gate (double positive for ST2 and Thy1.2) includes ILC2s isolated and sorted from lung tissue.

We have also conducted representative purification of other populations of ILC2s at week 2 and 3 (Appendix 1 Figure 2). We are including these here simply to demonstrate to the referee that the ILC2 we examine in our laboratory are routinely over 99% pure. Kindly see below:

Appendix 1 Figure 1:

Shown below are 2 additional representative purification of populations of ILC2s at week 2 and 3 in addition to that shown in Figure 1. The ILC2 we examine in this study are routinely over 99% pure. Gating strategy: Gating strategy for ILC2s from lungs were sorted by FACS as Lin⁻ ST2⁺ CD127⁺ CD90.2⁺ cells. ILC2s isolated from untreated animals (0.1% of total). Sequential gating strategy was based on cell size [small and non-granular cells, forward scatter (FSC) versus side scatter (SSC)], depletion of doublets (FSC vs. Trigger Pulse Width), and selection of CD45⁺ cells. Cells with lineage-related markers were rigorously depleted during isolation process and the resultant cell numbers at each stage of purification is shown. ILC2s isolated from IL-33-treated animals (0.3% of total). The final gate (double positive for ST2 and Thy1.2) includes ILC2s isolated and sorted from lung tissue. Purified Lin⁻ ST2⁺ CD127⁺ CD90.2⁺ ILC2s express GATA3, IL5 and IL13 and this is stable during culturing.

Additionally, though we did not find single marker molecules for individual subgroups of ILC2s, which is probably interesting to note by itself. However, we did find from our transcriptomic profiling data using the single cell sequencing, several subpopulations of ILC2s expressing unique combinations of marker molecules. For example, we find ILC2 that are equipped with T cell recruitment/attractant: Cxcl10, Ccr2 and Ltb, as well as pro-inflammatory and pro-survival signaling. Based on the presence of the large subtype (cluster 0+1), we propose that one of the mechanisms is through augmenting T cell migration and CTL differentiation at the tumour site, where ILC2s are expected to be present through IL33 signaling. *Type 2 Innate Lymphocytes Actuate Immunity Against Tumours and Limit Cancer Metastasis* Saranchova *et al.*, (2018) (<https://www.nature.com/articles/s41598-018-20608-6>) details a mechanism of action of ILC2s we propose during tumour development.

2. It is known that the number of ilc2 cells in peripheral blood is very small, and the number of cells immersed in tumor tissue should not be enough. How can you guarantee a certain number of ilc2 cells for testing?

While, we have removed the adoptive transfer experiments from the current manuscript and have focused on single cell analysis, we agree with the referee that the scarcity of ILC2s represented a barrier for utilizing these cells and to address this we recently published a paper describing how we now purify and expand ILC2s *in vitro* that is generally applicable: *Serum free culture for the expansion and study of type 2 innate lymphoid cells* de Lucia Finkel *et al.*, 2021 (<https://www.nature.com/articles/s41598-021-91500-z>).

Furthermore, in the present study, we overcame the scarcity of ILC2s in our data analysis by merging the week2 and week3 data in our analysis as one dataset to capture the changes induced by activation:

In the case of non-activated samples, ILC2 cell numbers were low, therefore we merged the two time points to obtain a more statistically robust dataset. Combining week2 and week3 datapoints allows us to look for significant changes that are specifically regulated by activation status.

In summary, we thank the referees for their comprehensive and insightful comments and believe they have significantly focused and improved our study which is the first description in the literature of the plasticity of ILC2s during tumour expansion.

Reviewers' comments:

Reviewer #1 (Remarks to the Author):

In this revised manuscript, the author addressed the issues that I concerned. They further demonstrated the tumor immune microenvironment and the interactional pattern between ILC2s and niche to against tumor. Furthermore, they proved the purification of ILC2s and detected the expression of TCR-related genes in ILC2s. However, there are few mistakes should be correct.

1. The text in Figure 2B was not legible.
2. In Figure 3A, the text in the heatmap should be removed.

Reviewer #2 (Remarks to the Author):

In this revised version of the article, the authors made a lot of changes but it is still very hard to read and many claims are not supported by the data. The figures are not properly organised, for instance, figure 4 is called before figure 2A and 3 and figure 2B is called before the 2A. It does not make sense. In figure 1, the two bottom pannels are completely ignored both in the text and the figure legend. The legend at the right of fig2B is not readable and there are two seven and two nine written.

P7 line 159, the authors talk about 2 clusters that have never been mentioned before in this version of the article, we are left to guess the ones they talk about. p7 line 162, figure 2C should be called.

P8 line 167, what are the biomarkers? are we supposed to guess again? Then we understand CD3 is a biomarkers but we do not know of which cluster and the other biomarkers remain unnamed.

Figure 3A, the words cluster 0 and cluster 1 appear on the heat map. It would have been interesting to see the expression of these genes for the other clusters as well.

p9 line 192/3, that claim is not sustained by data, the authors have to prove their statement, this type of suggestion can be discussed in the discussion only.

One of the major flaw of the article is the lack of controls. A single RNA seq on ILC2 from healthy mice should have been done to make sure the heterogeneity in indeed promoted by the tumor. Likewise, to state on the level of expression of genes Th1 related, a positive control is required, at least ILC1.

The meaning of APP is to be provided.

p10 line 222/223 the sentence is meaningless and line 233 the claim is not proven.

p11 line 246, we need to know the subtypes the authors are referring to.

Overall, the increase in the various genes seems to be anecdotal and for the most part are not convincing. The authors need to support their claims with functional data (cocultures, elisa, experiment with blocking antibodies...).

The figure legends are sometimes lacking informations and some titles are misleading (figure 8 for instance).

Reviewers' comments:

Reviewer #1 (Remarks to the Author):

In this revised manuscript, the author addressed the issues that I concerned. They further demonstrated the tumor immune microenvironment and the interactional pattern between ILC2s and niche to against tumor. Furthermore, they proved the purification of ILC2s and detected the expression of TCR-related genes in ILC2s. However, there are few mistakes should be correct.

Firstly, I would like to thank you for taking the time to re-review our revised manuscript.

1. The text in Figure 2B was not legible.

Apologies for that oversight. The text in Figure 2B has been replaced and is legible and it is now Figure 2A:

2. In Figure 3A, the text in the heatmap should be removed.

Figure 3A has now been moved to be figure 2B. The text in the heatmap has been removed:

Reviewer #2 (Remarks to the Author):

In this revised version of the article, the authors made a lot of changes but it is still very hard to read and many claims are not supported by the data. The figures are not properly organised, for instance, figure 4 is called before figure 2A and 3 and figure 2B is called before the 2A. It does not make sense.

Thank you for your observation, We have re-organized the manuscript so the figures are consecutively mentioned in the paper.

In figure 1, the two bottom panels are completely ignored both in the text and the figure legend. The legend at the right of fig2B is not readable and there are two seven and two nine written.

Apologies for that oversight.

Figure 1 has been redone to remove the bottom 2 panels as they are not relevant. We have only kept the relevant gating strategy as follows:

Figure 1. Gating strategy for isolation of ILC2s from lungs. The gating of a standalone experiment to illustrate the gating strategy, which is the same for all ILC2 isolations in the study. ILC2 cells were enriched with EasySep Mouse ILC2 Enrichment Kit first then sorted from Lungs of tumour bearing animals by FACS as Lin⁻ ST2⁺ CD127⁺ CD90.2⁺ cells. Sequential gating strategy is based on cell size [P1: small and non-granular cells, forward scatter (FSC) versus side scatter (SSC)], depletion of doublets (P2: SSC vs. Trigger Pulse Width), and selection of CD45⁺ cells. Cells with lineage-related markers were rigorously depleted during isolation process and the resulting cell numbers at each stage of purification is shown. The final gate (double positive for ST2 and Thy1.2) includes ILC2s isolated, enriched and sorted from lung tissue.

The text in Figure 2B has been replaced to be legible and it is now Figure 2A.

Furthermore one of the '7' and '9' that was written have been removed:

P7 line 159, the authors talk about 2 clusters that have never been mentioned before in this version of the article, we are left to guess the ones they talk about. p7 line 162, figure 2C should be called.

The 2 clusters that we were referring to are Cluster 0 and 1, However, no method that we attempted automatically combined the two clusters. The same goes to cluster 6 and 8.

Figure 2C has now been quoted in the location you suggested.

We have re-done the whole analysis and now each of the clusters are addressed individually.

P8 line 167, what are the biomarkers? are we supposed to guess again? Then we understand CD3 is a biomarkers but we do not know of which cluster and the other biomarkers remain unnamed.

The biomarker used for the experiment in Figure 3 is CD3e.

However, if the appendix Figure 2 which shows all the lineage profiles we conducted, biomarkers included CD3e, CD4, CD8A, CD11C, CD11D, and CD19. We just chose to include the lineage profile for CD3e in the text of the manuscript as it has the greatest importance for this experiment.

This has been made clear in the text.

For the lineage markers, they are in the material and methods. For the markers for specific clusters in the single cell analysis, the markers can be found in the Table 2 and 3b, they are now referred to as Gene of Interest (GOIs)

Thanks again for the comment.

Figure 3A, the words cluster 0 and cluster 1 appear on the heat map. It would have been interesting to see the expression of these genes for the other clusters as well.

Thank you for your observation. We have completely redone the figure and is now Figure 2B:

p9 line 192/3, that claim is not sustained by data, the authors have to prove their statement, this type of suggestion can be discussed in the discussion only.

Agreed, we have now moved this sentence to the discussion section of the manuscript.

Thanks for this comment.

One of the major flaw of the article is the lack of controls. A single RNA seq on ILC2 from healthy mice should have been done to make sure the heterogeneity in indeed promoted by the tumor. Likewise, to state on the level of expression of genes Th1 related, a positive control is required, at least ILC1.

We have repeated the experiment with a naïve, non-treated, non-activated group which is incorporated into the new analysis as the control. We did not include ILC1, but we included transcription factor markers to infer cell identity.

The meaning of APP is to be provided.

Sorry for that oversight. APP stands for antigen processing pathways, this has been clarified in the text.

p10 line 222/223 the sentence is meaningless and line 233 the claim is not proven.

Thank you, it has been amended to the following:

“Not only does CD74 play a crucial role in antigen presentation through MHC class II and cross-presentation through MHC class I by dendritic cells, it also functions as a receptor that can induce IL-8 production through NF- κ B activation upon interaction with migratory inhibitory factor (MIF)”

p11 line 246, we need to know the subtypes the authors are referring to.

Cluster 0 and 1. This has been added to the manuscript.

The average expression of *Klrg1* in total ILC2s synchronizes with this change by expressing the highest level in the naïve-ILC2 sample and lowest in the ta-activated ILC2s, resulting in a positive 0.35 log₂-fold change (log₂FC) in total naïve ILC2s compared to ta-ILC2s. Inflammatory ILC2s are known to be highly plastic towards a ILC3-like identity when cultured under Th17 conditions or exposed to infections. This is largely due to their basal level background ROR γ T expression (as observed in our naïve ILC2s, **Table 2**), and IL13, IL17A productions.

Overall, the increase in the various genes seems to be anecdotal and for the most part are not convincing.

The data was carefully analyzed by standard statistical test and determined to be statistically significant based on the following:

Statistics

Data were analyzed with R and Excel. Differential expression in single cell RNA-seq analysis is presented by logarithmic fold change, significance p is determined by Wilcoxon Rank Sum test, $p \leq 0.05$ was considered significant. A Student's t test was used for determining statistical significance between groups in ADT; $p \leq 0.05$ was considered significant.

We hope this response addresses the Referee's concerns.

The authors need to support their claims with functional data (cocultures, elisa, experiment with blocking antibodies...).

We understand the Referee's interest in these studies. We confirmed expression of protein markers and we are undertaking functional experiments that are the subject of another paper that is in revision.

The figure legends are sometimes lacking informations and some titles are misleading (figure 8 for instance).

Thank you for that observation. We have amended the title of Figure 8 (Which is now Figure 5) to: **Figure 5. The immunological plasticity of ILC2 cells has been defined by a new**

developmental program responsible for heightened type 1 immune signature, regardless of the stability of Th2 signatures.

As well as completely redone the figures and produced new figures, and amended the figure legends accordingly.

We hope this is satisfactory and we thank the referee for their comments.

REVIEWERS' COMMENTS:

Reviewer #2 (Remarks to the Author):

The authors did a great job at modifying their manuscript. It is much improved in my opinion.